

# Sedproxy: a forward model for sediment archived climate proxies

Andrew M. Dolman[1] and Thomas Laepple[1]

[1]Alfred Wegener Institute, Helmholtz Centre for Polar and Marine Research, Germany.

**Correspondence:** Andrew M. Dolman (andrew.dolman@awi.de)

**Abstract.** Climate reconstructions based on proxy records recovered from marine sediments, such as alkenone records or geochemical parameters measured on foraminifera, play an important role in our understanding of the climate system. They provide information about the state of the ocean ranging back hundreds to millions of years and form the backbone of paleo-oceanography.

However, there are many sources of uncertainty associated with the signal recovered from sediment archived proxies. These include seasonal or depth habitat biases in the recorded signal, a frequency dependent reduction in the amplitude of the recorded signal due to bioturbation of the sediment, aliasing of high frequency climate variation onto a nominally annual, decadal or centennial resolution signal, and additional sample processing and measurement error introduced when the proxy signal is recovered.

Here we present a forward model for sediment archived proxies that jointly models the above processes, so that the magnitude of their separate and combined effects can be investigated.

   Applications include the interpretation and analysis of uncertainty in existing proxy records, parameter sensitivity analysis to optimize future studies, and the generation of pseudo-proxy records that can be used to test reconstruction methods. We provide examples, such as the simulation of individual foraminifera records, that demonstrate the usefulness of the forward

model for paleoclimate studies. The model is implemented as a user-friendly R package, *sedproxy*, the use of which we hope will contribute to a better understanding of both the limitations and potential of marine sediment proxies to inform about past climate.

## 1 Introduction

Climate proxies are an imperfect record of the earth's past climate. Climate variations are encoded by geo- or bio-chemical

processes into a medium which survives, archived, until it is sampled and the physical or chemical signal decoded back into estimates of direct climate variables. For example, the ratio of magnesium to calcium in the shells (tests) of marine foraminifera varies with the temperature at which they calcify and thus encodes a temperature signal. Upon death, these shells (the carrier) sink to the ocean floor and become buried (archived) in the sediment. They can later be recovered from sediment cores and their Mg/Ca ratio measured. Using the modern day relationship between foraminiferal Mg/Ca and temperature, down-core

variations in the Mg/Ca ratio in foraminiferal tests can then be decoded back into an estimate of temperature variations back in time.



The climate signal is distorted and obscured at many points during the encoding, archiving and subsequent reading of a climate proxy, and these diverse sources of noise and error need to be taken into account when estimating the true past climate from proxy records. One way to develop, test, and improve our ability to reconstruct climate from proxies is to create mechanistic forward models. These models attempt to simulate the key processes on the entire path from the climate signal to

the reconstructed climate: from the encoding of the signal, its archiving in e.g. ice, sediments, wood or coral, recovery of the archived material, cleaning and processing of samples, measurement of the physical or chemical proxy, and its conversion back into units of climate variables such as temperature. Models that attempt to cover this entire process are known as proxy system models (PSMs) (Evans et al., 2013) and detailed PSMs have recently been proposed and implemented for oxygen isotope proxies archived in ice, trees, speleotherms and corals (Dee et al., 2015).

Climate proxies recovered from sediment cores are widely used to reconstruct past climate evolution on time-scales from centuries (Black et al., 2007) up to millions of years (Zachos et al., 2001). Several processes affecting the climate signal during recording, recovery and measurement have been described in the literature and analysed in specific studies. Examples include the influence of seasonal recording (Schneider et al., 2010; Leduc et al., 2010; Lohmann et al., 2013), the effect of bioturbation (Berger and Heath, 1968; Goreau, 1980), the sample size of foraminifera (Killingley et al., 1981; Schiffelbein and Hills, 1984)

and measurement uncertainty (Greaves et al., 2008; Rosell-Melé et al., 2001). Despite this body of knowledge, in practice these processes are often considered only in isolation, or not at all, when marine proxy records are interpreted, or when model-data comparisons are made.

The R package *sedproxy* provides a forward model for sediment archived climate proxies so that the above processes can be considered during study design, the interpretation of marine proxy records and when comparing models with data. *sedproxy* is

based on and expands the model described and used by Laepple and Huybers 2013 to explain differences in variance between Uk'37 and Mg/Ca based climate reconstructions. We first give an overview of the aspects of proxy creation that *sedproxy* can simulate. We then demonstrate how to use the package with a diverse series of use-cases. The source code for the specific version of *sedproxy* used to generate the examples used in this paper is contained in supplement S2, and the latest version of the code and R package are available on Bitbucket https://bitbucket.org/ecus/sedproxy.

## 25  2   Sediment archived proxy creation

The creation of a proxy climate record can be thought of as having three stages: sensor, archive and observation (Evans et al., 2013). Here we describe, for sediment archived proxy records, the key processes that occur in each of these stages and outline which of these are included in *sedproxy*.

### 2.1   Sensor stage

In the context of a climate proxy, a sensor is a physical, biological or chemical process that is sensitive to climate (e.g. temperature), and creates a measurable record of the climate signal. For example, the widths of tree growth rings are sensitive to temperature and water availability and are preserved in tree trunks (Evans et al., 2013). Our forward model can be used





for any proxy sensor that records water conditions and is then deposited and archived in the sediment. We consider here, as examples, two climate sensors: the Mg/Ca ratio in the tests of foraminifera, and the alkenone unsaturation index (Uk'37). Foraminifera are single celled protozoa that exude a calcite shell (test) in which a certain proportion of the calcium ions are substituted for magnesium. The ratio of Mg to Ca ions is dependent on the ambient temperature during the process of calcite

formation, and thus the Mg/Ca ratio in foraminiferal tests acts as a proxy for temperature during their creation (Nürnberg et al., 1996). Similarly, alkenones are a class of large organic molecules synthesised by some Haptophyte phytoplankton species. The proportion of unsaturated carbon to carbon bonds in the synthesised molecules is temperature dependent and thus the relative unsaturation of alkenone molecules found in sediments can be used as a proxy for temperature (Prahl and Wakeham, 1987). Secondary effects such as the effect of salinity on the Mg/Ca of foraminifera (Hönisch et al., 2013), or nutrient availability on

the Uk'37 recorded by the alkenone producers (Conte et al., 1998), might further effect the recorded proxy signal.

So as to be applicable to a wide range of climate sensor types, we do not explicitly model the encoding process for specific sensors. Other tools have been developed that do this, e.g. FIRM for foraminiferal $\delta^{18}$O (Fraass and Lowery, 2017), and could be used to pre-process the input climate signal. Rather we include a general method for adding error due to uncertainty in the estimated proxy calibration.

### 2.1.1  Seasonal and habitat bias in the sensor

One source of uncertainty common to most climate proxies is a bias towards recording the climate during periods of the year when the proxy generating process is most active (Mix, 1987). Both the foraminifera and the alkenone producing haptophytes have growth rates, abundances and rates of export to the sediment that vary predictably throughout the year (Jonkers and Kučera, 2015; Leduc et al., 2010; Uitz et al., 2010), and hence bias these proxies towards recording the climate during their

respective periods of peak production and export. Furthermore, the proxy creating organisms do not necessarily live at and record the surface of the ocean. The producers of alkenones are restricted to the photic zone and thus are thus close to the surface. However, for foraminifera, the preferred habitat depth and the depth at which their shells calcify is strongly species dependent and can vary from close to the surface, to the thermocline or deeper (Fairbanks and Wiebe, 1980; Kretschmer et al., 2017). Therefore, the recorded temperature will not necessarily reflect the sea surface temperature (Jonkers and Kučera,

2017). Whether or not these biases represents an error will depend on how the resulting proxy record is interpreted. However, even when a proxy is interpreted as representing a particular season or depth habitat, the season and depth that a given proxy represents will rarely be known with certainty.

### 2.2  Archive stage

After the creation of proxy carriers such as foraminiferal shells or alkenone molecules, a proportion of these are exported to

and buried in the sediment. We assume here that this process is local and ignore the potential for lateral transport of the proxy material in the water column or at the sediment surface.



### 2.2.1 Bioturbation

The upper few centimetres of marine sediments are typically mixed by burrowing organisms down to a depth of around 2-15 cm (Boudreau, 1998, 9.8 ± 4.5 cm (1 SD)) (Teal et al., 2010; Trauth et al., 1997, 8.37 ± 6.19 cm), although laminated sediments absent of bioturbation do exist. Marine sediment accumulation rates vary over many orders of magnitude (Sadler, 1999; Sommerfield, 2006) but rates at core locations used for climate reconstructions are typically of the order 1-100 cm ka$^{-1}$. Thus, bioturbation can mix and smooth the climate signal over a period of many hundreds of years and has a strong effect on the effective temporal resolution that can be recovered from a sediment archived proxy (Anderson, 2001; Goreau, 1980).

Other processes occurring during the archive stage may influence the proxy, for example preferential dissolution of Mg/Ca in foraminiferal shells (Barker et al., 2007; Rosenthal and Lohmann, 2002; Mekik et al., 2007) and preferential degradation of Uk'37 (Hoefs et al., 1998; Conte et al., 2006). We assume here that these effect are minimal, or would be spotted during sample processing (e.g. dissolution of Mg/Ca), and the signal is preserved.

## 2.3 Observation stage

### 2.3.1 Aliasing of inter- and intra-annual climate variation

During the observation phase, samples of sediment are taken at intervals along a core and material is recovered in which the proxy signal has been encoded. For proxies embedded in the tests of foraminifera, this is typically a relatively small sample of about 10-30 individuals. Due to bioturbation, these individuals will be a mixed sample that integrate the climate signal over an extended time period; however individual planktonic foraminifera live for a period of only 2-4 weeks (Bijma et al., 1990; Spero, 1998) and hence each encodes climate at an approximately monthly resolution. Therefore, if a measurement is made on a sample containing 30 individuals mixed together from a period of 100 years, the resulting value is a noisy 100-year mean and hence inter- and intra-annual scale climate variation is aliased into the nominally centennial-resolution proxy record (Laepple and Huybers, 2013; Schiffelbein and Hills, 1984). This effect may be particularly strong for high latitude cores where the seasonal temperature cycle is large. However the stronger the seasonal climate cycle, the more likely an organism is to grow preferentially during a specific season (Jonkers and Kučera, 2015), and thus aliasing will be reduced, while seasonal bias is increased. For organic proxies such as Uk'37, samples comprise many thousands of molecules and aliasing is likely a minor issue, although clustering in sediment export and distribution is possible (Wörmer et al., 2014).

### 2.3.2 Other non-climate variability: inter-individual variation, cleaning/processing and instrumental error.

The measurement of proxy values on material recovered from sediment cores will necessarily involve some amount of error. In particular, foraminiferal tests need to be cleaned prior to Mg/Ca measurements and this is an imprecise process. Too little cleaning risks leaving Mg rich mineral phases (Barker et al., 2003), too much may bias the Mg/Ca downwards. Some cleaning, processing and measurement errors will be independent between samples while others may be correlated, for example due to differences between labs (Greaves et al., 2008). In addition to measurement error, there will also be inter-individual variation




between foraminifera in their recording of the same climate signal. For example, test Mg/Ca ratios vary between individual foraminifera even when grown under identical conditions (e.g., Dueñas-Bohórquez et al., 2011). Similar inter-individual variation, or "vital effects", also occur for $\delta^{18}$O (Schiffelbein and Hills, 1984).

An additional sampling artefact is created due to the need to pick individual foraminiferal tests, or extract Uk'37, from a slice of a sediment that cannot be infinitely thin. Therefore, even in the absence of bioturbation, this material will cover a time period determined by the sedimentation rate and layer thickness. Foraminifera are typically picked from 1-2 cm thick sediment layers, provided enough individuals can be found.

## 3 Implementation

*sedproxy* takes as input an assumed "true" climate signal, which may come from a climate model or instrumental readings, and returns a simulated proxy value for each of a set of requested timepoints. Returned values are in the same units as the input climate signal, which may be either temperature or proxy units. All required parameters and input variables are shown in Table 1, and described in the following paragraphs together with an overview of the model implementation.

### 3.1 Input climate matrix ("clim.signal")

The input climate signal is required at monthly resolution in order to be able to simulate seasonal biases in the recording process and noise aliased from monthly climate variation. It is useful, in the implementation, to view this climate signal as a years by months matrix $C_{y,m}$ where $y$ are the years and $m$ are the 12 months. To include habitat effects, e.g. foraminiferal depth habitats, this matrix can be extended to have 12 x $z$ columns, where $z$ is the number of discrete habitats $h$ to be included. In this case a depth resolved input climate signal would be required. For simplicity we consider only sea surface signals in the examples presented here.

$$\begin{pmatrix} C_{y_1,m_1,h_1} & C_{y_1,m_2,h_1} & \cdots & C_{y_1,m_{12},h_z} \\ C_{y_2,m_1,h_1} & C_{y_2,m_2,h_1} & \cdots & C_{y_2,m_{12},h_z} \\ \vdots & \vdots & \ddots & \vdots \\ C_{y_n,m_1,h_1} & C_{y_n,m_2,h_1} & \cdots & C_{y_n,m_{12},h_z} \end{pmatrix}$$

### 3.2 Weights matrix

While conceptually *sedproxy* modifies the climate signal according to a sequence of sensor, archive and observation processes, in practice the value of the simulated proxy at a given timepoint is calculated in a single step as the mean of a weighted sample from the original climate signal, plus some independent error term. For each requested timepoint, a matrix of weights, $W$, is constructed which determines the region of the original climate signal that will be sampled and the probability of sampling any particular value.





The weights matrix $W$ is the product of a column vector of annual weights, $w_y$, which depend on bioturbation, and a row vector of habitat weights, $w_{mh}$, which depend on the seasonality and potentially the depth habitat of the proxy recording process.

$$W = \begin{pmatrix} w_{y_1} \\ w_{y_2} \\ \vdots \\ w_{y_n} \end{pmatrix} \begin{pmatrix} w_{m_1 h_1} & w_{m_2 h_1} & \cdots & w_{m_{12} h_z} \end{pmatrix} = \begin{pmatrix} w_{1,1} & w_{1,2} & \cdots & w_{1,12z} \\ w_{2,1} & w_{2,2} & \cdots & w_{2,12z} \\ \vdots & \vdots & \ddots & \vdots \\ w_{n,1} & w_{n,2} & \cdots & w_{n,12z} \end{pmatrix}$$

### 3.2.1 Habitat weights (season and depth habitat of proxy production and export) ("proxy.prod.weights")

The habitat weights, $w_{mh}$, are given by a user defined vector defining the seasonality and potentially the depth habitat of the proxy recording process. It has the same length as the number of columns in the input climate signal. In this case where we use monthly sea surface temperature and ignore depth habitats, the habitat weights vector has 12 values. Currently the season and depth habitat in the recording (but not necessarily the climate) is assumed to be invariant over time.

### 3.2.2 Annual weights (bioturbation)

For simplicity, *sedproxy* assumes complete mixing within the bioturbated layer, a constant sedimentation rate in the region of each sampled timepoint, and a constant concentration of the proxy carrying material. Under these assumptions, the origin (pre-bioturbation) of material recovered from a given focal depth is described by the impulse response function Eq. (1) (Berger and Heath, 1968). This function is equivalent to an exponential probability density function, with mean equal to the focal depth and standard deviation equal to the bioturbation depth divided by the sedimentation rate. The value of a proxy measured on material recovered from a given depth can thus be viewed as a weighted mean of material originally deposited over a range of depths, with weights given by Eq. (1) (Fig. 1).

In this model, the probability that a particle found at a given focal depth was mixed down from a distance greater than the bioturbation depth, $\delta$, is zero. Theoretically, particles can have been brought up from any distance below the focal depth, but for computational reasons the annual weights vector is restricted to a distance of three bioturbation depths below the focal horizon; this region contains 99% of the mass of the impulse response function. By assuming a locally constant sediment accumulation rate, $\alpha$, around each focal point, and a fixed bioturbation depth, $\delta$, the bioturbation function can be expressed in units of time rather than space/depth.

$$wy_t = \begin{cases} \frac{\alpha \cdot e^{\lambda y_f - \lambda y_t - 1}}{\delta} & y_t - y_f + \frac{\delta}{\alpha} \geq 0 \\ 0 & y_t - y_f + \frac{\delta}{\alpha} < 0 \end{cases} \tag{1}$$

where:

$\alpha$ = sediment accumulation rate in cm a$^{-1}$



$\delta$ = bioturbation depth in cm

$\lambda = \frac{\alpha}{\delta}$

$y_f$ = the focal year, and

$x = y_t - y_f + \frac{\delta}{\alpha}$

To account for the fact that foraminiferal tests are collected, or Uk'37 extracted, from a layer of sediment of a certain thickness ("layer.width"). The bioturbation function is convolved with a uniform probability density function with a width equal to the layer thickness.

### 3.2.3   Summing or sampling

For proxies such as foraminiferal Mg/Ca, where typically a small number of foraminiferal tests ( 30) are cleaned and measured
for each depth/timepoint in a sediment core, the proxy at time $t$, $Pr_t$, is the mean of a random sample of $N$ elements of the input climate matrix $C$, with the probability that a particular element is sampled given by the weights matrix $W$, plus some some independent error term $\varepsilon$.

$$Pr_t = \frac{1}{N} \sum_{i=1}^{i=N} \{C^{(i)}, W^{(i)}\} + \varepsilon \tag{2}$$

For proxies such as Uk'37, it is assumed that there are effectively infinite samples taken for each timepoint at which the
proxy is evaluated. In this case the proxy at time $y$ $Pr_t$ is the sum of the element-wise product of the climate and weights matrices.

$$Pr_t = \sum (C \circ W) + \varepsilon \tag{3}$$

### 3.3   Independent error term ("meas.noise")

The error term $\varepsilon$ is added as an independent Gaussian random variable with mean $\mu = 0$, and standard deviation $\sigma$ dependent
on the proxy type. For foraminiferal Mg/Ca we use $\sigma = 0.46$, for Uk'37 $\sigma = 0.25$ (Laepple and Huybers, 2013). These errors represent not just instrumental measurement error, which is typically much smaller than the errors quoted here, but also included error introduced during, e.g. the cleaning of foraminiferal tests and other non-climate variability such as the inter-individual variability of Mg/Ca in foraminifera.

$$\varepsilon \sim \mathcal{N}(\mu, \sigma) \tag{4}$$

### 3.4   Replication

Multiple replicate proxy records can be simulated with a single set of parameters. Due to the stochastic sampling of monthly temperatures and the random noise term replicates will not be identical. An additional random bias can be added to each





replicate simulated proxy record. This bias is drawn from a Gaussian distribution with mean = 0 and a user definable standard deviation ("meas.bias" defaults to 0). This bias will be constant for all points in a given replicate and can be used to include uncertainty in the proxy calibration, or inter-lab variation in analytical results.

## 4   Using *sedproxy*

5   To illustrate the use of *sedproxy* we provide here a number of simple examples together with the R code to execute them.

### 4.1   Example 1: A foraminiferal Mg/Ca pseudo-proxy record for sediment core MD97-2141

In this first example, we demonstrate how to simulate an already measured proxy record as closely as possible. We use the foraminiferal Mg/Ca based temperature reconstruction for sediment core MD97-2141 (Table 2) in the Sulu Sea (Rosenthal et al., 2003).

10   As an input climate signal we take the monthly sea surface temperature output from the TraCE-21ka "Simulation of Transient Climate Evolution over the last 21,000 years" (Liu et al., 2009), using the grid cell closest to core MD97-2141.

The seasonality of *Globigerinoides ruber*, the foraminifera for which test Mg/Ca ratios were measured, is taken from the dynamic population model PLAFOM (Fraile et al., 2008) (Fig. 2a). Sediment accumulation rates were estimated from the depth and age data associated with core MD97-2141 and provided in the supplemental data to Shakun et al 2012. These data

15   are included in the *sedproxy* R package as example data and are also used in the later examples.

The function `ClimToProxyClim` is used to forward model a proxy record from an assumed climate. We request values of the proxy at the timepoints of the observed proxy. Descriptions of all the function arguments can be found in Table 1. Or from the R console type `?ClimToProxyClim` to see the help page.

```
library(sedproxy)
```





```
# Reverse matrix so that top row is most recent year,
# also convert from Kelvin to °C
N41.t21k.climate.in <- N41.t21k.climate[nrow(N41.t21k.climate):1, ] - 273.15

# Convert matrix to a ts object and set start to most recent year,
# in this case -39 (1989 in years "before" 1950)
N41.t21k.climate.in <- ts(N41.t21k.climate.in, start = -39)

# Set seed of random number generator so that the results are reproducable.
set.seed(20170824)

# Call the forward model
Mg_Ca.30 <- ClimToProxyClim(
  clim.signal = N41.t21k.climate.in,
  timepoints = N41.proxy$Published.age,
  sed.acc.rate = N41.proxy$Sed.acc.rate.cm.ka,
  smoothed.signal.res = 1,
  proxy.prod.weights = N41.G.ruber.seasonality,
  meas.noise = 0.46,
  n.samples = 30, n.replicates = 1)
```

In addition to the estimated final proxy timeseries, *sedproxy* calculates and returns the unobserved intermediate stages of proxy creation to assist in the interpretation of the simulated proxy. We provide a plotting function `PlotPFMs` which will display the output from `ClimToProxyClim`, together with an observed proxy record if this is added to plotting data. `PlotPFMs` returns a ggplot object that can be customised using the standard ggplot functions (Wickham, 2009).

```
plot.dat <- Mg_Ca.30$everything
```



```
# Rescale timepoints to ka for plotting
plot.dat$timepoints <- plot.dat$timepoints / 1000

# Add observed proxy record
obs.proxy <- data.frame(timepoints = N41.proxy$Published.age / 1000,
                        value = N41.proxy$Published.temperature,
                        stage = "observed.proxy", replicate = 1)
plot.dat <- bind_rows(obs.proxy, plot.dat)

p.MD97_2141 <- PlotPFMs(plot.dat, stage.order = "seq") +
  facet_wrap(~stage, labeller = as_labeller(stage.labels))  +
  scale_x_continuous("Age [ka BP]") + theme(legend.position = "none")
p.MD97_2141
```

Fig. 3 shows the forward modelled Mg/Ca proxy record for core MD97-2141 (5), together with the input climate signal smoothed to centennial resolution (1), the intermediate stages of proxy creation (2-4), and the observed proxy reconstruction as published in Rosenthal et al. 2003. Although the observed (*) and forward modelled (5) proxy records appear to have similar variance, the simulated bioturbation first removes most features of the input climate signal before the aliasing and noise term

increase the variability again. In this example, the median sediment accumulation rate is 25.6 cm ka$^{-1}$, which, assuming a bioturbation depth of 10 cm, corresponds to an expected standard deviation in the ages of individual foraminifera recovered from a single depth of 390 years. Trends remain visible at temporal resolutions of approximately 2 ka and greater, as does a single centennial-to-millennial scale feature present in the input climate signal at around 12.5 ka BP.

The combination of the seasonal temperature cycle present in the monthly TraCE-21ka simulation, and the seasonality of

*G.ruber* taken from Fraile et al. 2008, shifts the forward modelled proxy by about -0.26 °C (Fig. 3, 2-3). This shift varies from -0.29 to -0.16 °C depending on the strength of the seasonal cycle, which changes due to the variations in the orbital parameters.

The single centennial-to-millennial scale feature still visible in the bioturbated signal at 12.5 ka BP is obscured first by the effects of aliasing of annual and intra-annual variance, dominated by the seasonal climate cycle, onto the proxy record due to relatively small number of foraminifera contributing to each proxy data point. Further measurement error erases any trace of

these centennial-to-millennial scale features in the final forward modelled proxy; only multimillenial and greater scale trends remain visible.

The resolution of features that can be seen in the final forward-modelled proxy is consistent with Rosenthal et al. 2003's interpretation of the observed Mg/Ca proxy, from which they estimate the LGM-Holocene temperature increase, but find no other significant features. However, the features visible in a forward modelled proxy are of course dependent on both the input

climate signal - in this case the TraCE-21ka simulation - and parameter values used in the proxy simulation.





## 5   Example 2: Influence of the number of foraminifera per sample

To examine the influence of the number of individual foraminifera per timepoint on the uncertainty due to seasonal aliasing,
we simulate two artificial Mg/Ca records with 1 and 30 individual foraminifera per sample. For comparison, we also simulate
a Uk'37 record, for which the sample size per timepoint is assumed to be infinite. For simplicity we assume that alkenones are
5  produced uniformly throughout the year.

```
Mg_Ca.1 <- ClimToProxyClim(
  clim.signal = N41.t21k.climate.in,
  timepoints = N41.proxy$Published.age,
  sed.acc.rate = N41.proxy$Sed.acc.rate.cm.ka,
  proxy.prod.weights = N41.G.ruber.seasonality,
  meas.noise = 0.46, n.samples = 1)

Uk37 <- ClimToProxyClim(
  clim.signal = N41.t21k.climate.in,
  timepoints = N41.proxy$Published.age,
  sed.acc.rate = N41.proxy$Sed.acc.rate.cm.ka,
  meas.noise = 0.25, n.samples = Inf)
```

The output from these three runs of the model is shown in Fig. 4. For brevity, code to generate the figure and perform the
simulation with 30 individuals is not shown here but complete code for all examples is provided as supplementary material.

## 6   Example 3: Correlation between two proxy types.

*sedproxy* can be used to explore the expected correlation between pairs of proxy records. Here we correlate Mg/Ca and Uk'37
10  based proxies generated for the same hypothetical sediment core. Records from different locations could be compared by
supplying a different input climate matrix for each site.

To emphasise the potential effect of contrasting proxy seasonality on the correlation between two records we use hypothetical
seasonal weights. The Uk'37 proxy is again assumed to have a constant production with no seasonality, while production of
the Mg/Ca proxy is heavily weighted towards August and September.

15  We again use the same TRaCE-21ka input climate but for simplicity we use a constant sedimentation rate and request proxy
values at equally spaced timepoints. One thousand replicate proxy records are simulated of each type.

```
Uk37.reps <- ClimToProxyClim(
```




```
  clim.signal = N41.t21k.climate.in,

  timepoints = seq(100, 21000, by = 1000),

  sed.acc.rate = 25, proxy.prod.weights = rep(1/12, 12),

  meas.noise = 0.25,

  n.samples = Inf,  n.replicates = 1000)

MgCa.reps <- ClimToProxyClim(

  clim.signal = N41.t21k.climate.in,

  timepoints = seq(100, 21000, by = 1000),

  sed.acc.rate = 25,

  proxy.prod.weights = c(0, 0, 0, 0, 0, 0, 0.2, 0.7, 1, 0.6, 0, 0),

  meas.noise = 0.46,

  n.samples = 30, n.replicates = 1000)

proxies <- bind_rows("Mg/Ca"=MgCa.reps$everything,

                     "Uk'37"=Uk37.reps$everything,

                     .id = "Proxy")

proxies <- filter(proxies, stage == "simulated.proxy")
```

The Mg/Ca based artificial records show greater variance than Uk'37 due to a combination of aliasing caused by the finite number of foraminiferal tests and an assumption of higher measurement error (Fig. 5). In addition to a mean offset between the two proxy types, the hypothetical Mg/Ca proxy shows a much stronger glacial-interglacial transition because the effect of the bias towards recording summer climate increases when the amplitude of the seasonal cycle is larger and this was maximal at around 10 ka BP.

Fig. 6 shows the distribution of correlations between replicated pairs of hypothetical Mg/Ca, Uk'37, and Mg/Ca-Uk'37 records, calculated over both the past 10k years (Holocene), and the past 21k years which include the de-glaciation. Over the Holocene, the average correlation between simulated pairs of proxy records is low, even for pairs of the same proxy type. The average correlation between Mg/Ca and Uk'37 proxy records is even negative, due to the simulated warming annual mean temperature, sampled by the Uk'37 record, but slightly cooling summer temperature sampled here by the hypothetical summer growing foraminifera. Similar contrasting trends have been observed between real Mg/Ca and Uk'37 records over the Holocene (Leduc et al., 2010). Correlations between Uk'37 Uk'37 pairs are slightly higher than those between Mg/Ca pairs, due to the lower measurement noise and lack of aliasing we assume for Uk'37. When the proxy records include a large climate transition, such as the deglaciation between 21ka BP and 10ka BP, correlations between all pairs become high.




## 7    Example 4: Individual Foraminiferal Analysis

In individual foraminiferal analysis (IFA), the population statistics (e.g. standard deviation or range) of proxy values measured on individual foraminifera recovered from the same depth, are used to infer changes in climate variability - such as changes in the El Niño Southern Oscillation (ENSO) system (e.g., Koutavas and Joanides, 2012; Killingley et al., 1981), or changes in the amplitude of the seasonal cycle (e.g., Ganssen et al., 2011; Wit et al., 2010). *sedproxy* can be used to simulate IFA by setting 'n.samples = 1' and 'n.replicates' to the number of individuals measured per timepoint.

Motivated by the study from Scussolini et al. 2013, which examined changes in the IFA distribution of $\delta^{18}$O during the penultimate deglaciation, we simulate a case study that demonstrates the effect of bioturbation on the IFA distribution and choose parameter values resembling this study. The sedimentation rate is set to 1.3 cm ka$^{-1}$, we simulate 20 foraminiferal tests for the IFA analysis, 45 foraminiferal tests for the bulk measurements and assume a measurement noise of 0.1 ‰ $\delta^{18}$O for the IFA and the bulk measurements. To mimic the reconstructed climate signal of Scussolini et al. (2013), we assume a climate transition from 0.4 ‰ at 190 ka BP, to 2.6 ‰ at 90 ka BP with the shape of a logistic function. Finally, we add stochastic climate variability following power law scaling with slope = 1 (Laepple and Huybers, 2014), and variance 0.15 and sinusoidal seasonal variations with an amplitude of 0.5 [‰ $\delta^{18}$O]. These choices are partly arbitrary but reproduce similar IFA and bulk variance as those shown in Scussolini et al. 2013 (Fig. 7).

At the measured sediment accumulation rate of 1.3 cm ka$^{-1}$ and with an assumed bioturbation depth of 10 cm, the expected standard deviation in ages of material found at a given depth is approximately 7900 years. Thus bioturbation mixes material across the deglaciation, so that samples with a mean age of between 110 and 140 ka BP contain a mixture of glacial and inter-glacial material, and hence show a higher standard deviation in $\delta^{18}$O, with a peak at around 135 ka BP (Fig. 8). This demonstrates that bioturbation can have a significant effect on the IFA distribution in low sedimenation rate settings.

## 8    Discussion and conclusions

We present a forward model for the simulation of marine sediment based proxy records from climate data. We choose to include the main well constrained processes affecting sedimentary signals while keeping it general enough to be usable for a large set of problems. The *sedproxy* model is implemented as a user-friendly R package in an open-source framework (R Core Team, 2017).

Our forward model combines and extends the work of many previously published studies and models concerning the formation of sedimentary records. For example, several prior studies have investigated the effect of seasonality and/or depth habitat on the recorded proxy signal (e.g., Leduc et al., 2010; Liu et al., 2014; Lohmann et al., 2013; Schneider et al., 2010). In addition to the effect on the signal evolution and trends considered in these studies, *sedproxy* includes the effect on proxy variability caused by the finite sample size in combination with the habitat range.

For the specific application of interpreting the variability of individual foraminifera (IFA), our model bears some similarities with INFAUNAL (Thirumalai et al., 2013); however, while INFAUNAL was designed to test the sensitivity of IFA to the sea-




sonal cycle and inter-annual variability, and therefore includes a specific analysis on the simulated IFA distributions, *sedproxy* is more general and also includes the effects of bioturbation, such as shown in Example 4.

Several previous studies have examined how bioturbation reduces the amplitude of the recorded signal, and in combination with noise puts a limit on the temporal resolution of climate events that can be resolved in proxy records (Anderson, 2001; Goreau, 1980), and tools have been developed to model bioturbation (Trauth et al., 1997). While being simpler than some of these approaches, the combination in *sedproxy* of bioturbation with the other effects, such as the seasonal aliasing or the measurement error, allows the interaction between these effects to be investigated.

The relative importance of bioturbation, seasonal biases, aliasing and other noise sources will vary according to the physical characteristics of the sediment core (e.g. sediment accumulation rate), the length of the record, the amplitude of the seasonal cycle, and the amplitude of the signal that is being reconstructed (e.g. a glacial-interglacial transition vs. ENSO). Most importantly, the type of information that is sought from the proxy record will determine whether these errors are processes are important. By jointly simulating the major processes affecting the sediment record, *sedproxy* allows these to be considered together.

## 8.1 Applications

*sedproxy* has many potential applications in paleoclimate research, not limited to those in the examples given above. It can serve as a forward model to create more realistic surrogate records that can be used to test climate field reconstruction methods (e.g., Smerdon et al., 2011) and it can further act as a forward model for inversion based climate reconstructions methods for example using Bayesian hierarchical models (Tingley and Huybers, 2009) or data assimilation schemes (e.g., Klein and Goosse, 2017). Importantly, it allows quantification of the full uncertainty of proxy records related to the processes included in the model. By providing an ensemble of surrogate (pseudo) proxy realizations, rather than single error values, the full temporal structure of the uncertainty can be characterized. Proxy uncertainty can be determined as a function of time-scale, thus separating uncertainties affecting long-term means or time-slices, such as the seasonal recording effects, from temporarily independent noise, such as that caused by aliasing of the seasonal cycle. This enables more quantitative comparisons to be made between climate models and proxy data than would classical direct comparison.

The ability to analyse intermediate stages of the simulated proxy (see example 1) allows the effects of different error sources to be evaluated. Used in this way, *sedproxy* can help optimize and test sampling strategies for sediment cores by evaluating the effect of e.g. the sample thickness, number of foraminifera or analytical uncertainty on the final record. This information can be used to improve the design of studies and to test, prior to a study, whether signals of interest such as centennial scale climate variations could theoretically be resolved by the proxy record.

## 8.2 Caveats and current limitations

While being relatively simple and general, there are inherent caveats to the present forward modelling approach: *sedproxy* does not currently include a complex sensor model - the input climate signal and output proxy signal have the same units, for example temperature. While this allows for general application to different proxies archived in marine sediments, it does not



account for the uncertainties created during the process of encoding the signal in the proxy material. To overcome this, the input climate signal can be converted to proxy units prior to running the forward model. Any given sensor model could be used, from simple linear or exponential regression to more complex process based sensor models. A back-transformation can then be applied to the generated pseudo-proxy records, which itself might model uncertainty by varying the parameters of the

calibration.

In its current version, *sedproxy* can be used to simulate mean shifts in the recorded climate signal due to seasonality or depth habitat preferences, but not the effects of climate dependent shifts in timing and depth habitat (i.e. habitat tracking, Jonkers and Kučera, 2017), which will likely damp the recorded changes (Fraile et al., 2009). In addition to shifting the seasonal bias in the recorded climate signal, the absolute concentration of the carrier (e.g. foraminfera species) can also change over time

in response to climate and this would interact with bioturbation, potentially shifting the apparent timing of climate transitions (Bard et al., 1987; Hutson, 1980). Currently, a constant concentration of the signal carrier is an assumption of *sedproxy*. Future work will enable climate dependent shifts in habitat and abundance to be modelled by implementing a parametrized response of proxy abundance and export to climate variables (Mix, 1987; Schmidt and Mulitza, 2002; Kretschmer et al., 2017; Jonkers and Kučera, 2017; Roche et al., 2017). An alternative option is to couple *sedproxy* to the output of ecological models that

explicitly resolve the population dynamics of the proxy carrier, such as foraminifera population models (Fraile et al., 2008; Lombard et al., 2011).

For simplicity, we implemented a minimal physical model for bioturbation that assumes a completely mixed bioturbated layer, with a sharp cut-off to zero mixing below this layer (Berger and Heath, 1968). However, the general effect of bioturbation should also apply under conditions of incomplete mixing and the code could easily be modified to use a more complex

bioturbation model (e.g., Guinasso and Schink, 1975; Steiner et al., 2016) to generate the weights used to sample the input climate signal. We note that when sedimentation rates are low relative to mixing rates, more complex mixing models converge to the simple box type model that we employ here (Matisoff, 1982).

Our model further assumes a constant bioturbation depth over time, as the bioturbation depth is generally not known for each setting and cannot easily be reconstructed down-core. Bioturbation depth may be related to productivity and sedimentation

rate, but its predictability for a given core seems to be low (Trauth et al., 1997). The recent development of radiocarbon measurements on small samples (Wacker et al., 2010) might allow the extent of bioturbation to be constrained using replicate measurements from individual depth layers and such information could easily be included in *sedproxy*.

In contrast to some other proxy system models that have been proposed for corals, ice, trees and speleothems (e.g., Dee et al., 2015), *sedproxy* currently does not explicitly include the depth to age conversion and thus does not account for chronological

uncertainty. In future studies, radiocarbon could be included in the forward modelling and thus the link between finite sample size, bioturbation and chronological uncertainty could be included.

Finally, in our examples we assumed that the climate signal recorded is that of the water column directly above the core location. There is evidence that this is not always the case, especially in dynamic regions and at drift deposits (Mollenhauer et al., 2003; van Sebille et al., 2015). This effect could be included by providing the non-local climate information as input to

the forward model.





While *sedproxy* largely relies on well understood processes that have been previously described in the literature, there is a strong need to refine this and other proxy system models and to confront them with observational data. For this purpose, more systematic multiproxy studies comparing independent proxies from the same archives (e.g., Ho and Laepple, 2016; Laepple and Huybers, 2013; Weldeab et al., 2007; Cisneros et al., 2016) would be useful. Studies analysing replicability inside

and between sediment cores in analogue to studies for ice and coral based proxies (DeLong et al., 2013; Smith et al., 2006; Münch et al., 2016) would allow better constraint of the sample error parameter. Likewise, further investigation of potentially important processes occurring during the preservation of archived proxy signals (e.g., Münch et al., 2017; Zonneveld et al., 2007; Kim et al., 2009) would allow these to be included in proxy system models. Finally, modern core-top studies of individual foraminifera distributions (e.g., Haarmann et al., 2011) would allow further testing of the assumption that there is a direct link

between proxy variability and climate variability.

We hope that this tool will be useful to the paleoclimate research community and we hope that it can provide a starting point for a more complete future proxy system model for sediment proxies. We invite external contributions via the Bitbucket repository, https://bitbucket.org/ecus/sedproxy.

*Code and data availability.* The forward model *sedproxy* is implemented as an R package and its source code is available from the pub-

lic git repository at https://bitbucket.org/ecus/sedproxy. The R package also contains the data needed for the examples. R code to run all the examples in this manuscript is contained in supplement S1. Source code for the specific *sedproxy* version used to create the examples in this manuscript is contained in supplement S2. An interactive example showing the main features of *sedproxy* can be accessed at https://limnolrgy.shinyapps.io/sedproxy-shiny/

*Competing interests.* The authors declare that they have no conflict of interest.

*Acknowledgements.* This work was supported by German Federal Ministry of Education and Research (BMBF) as Research for Sustainability initiative (FONA); www.fona.de through Palmod project (FKZ: 01LP1509C). T. Laepple was supported from the European Research Council (ERC) under the European Union's Horizon 2020 research and innovation programme (grant agreement no. 716092) and the Initiative and Networking Fund of the Helmholtz Association grant VG-NH900. We thank Guillaume Leduc for suggesting example uses of the forward model and Jeroen Groeneveld, Michal Kučera and Lukas Jonkers for helpful comments on the manuscript and advice during

development of the ideas.



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





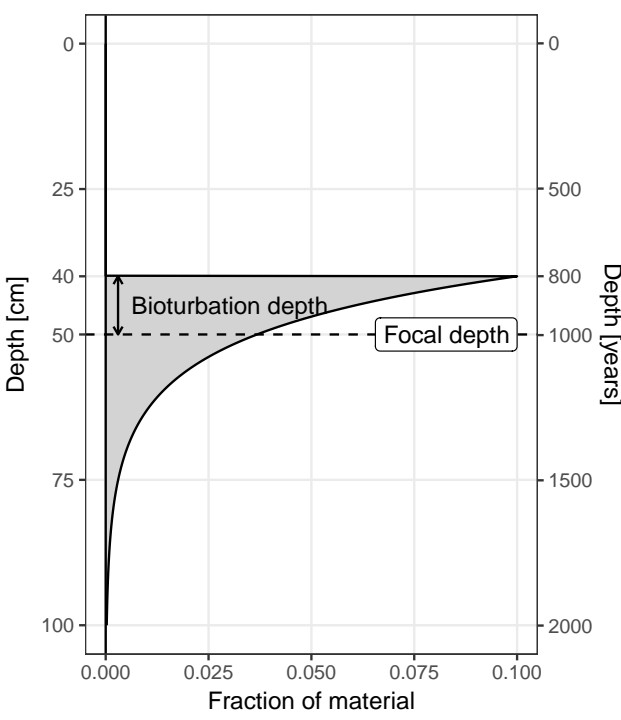

**Figure 1.** The origin of material archived at a focal core depth of 50 cm. In this example the bioturbation depth is 10 cm, and the sediment accumulation rate is 50 cm ka$^{-1}$





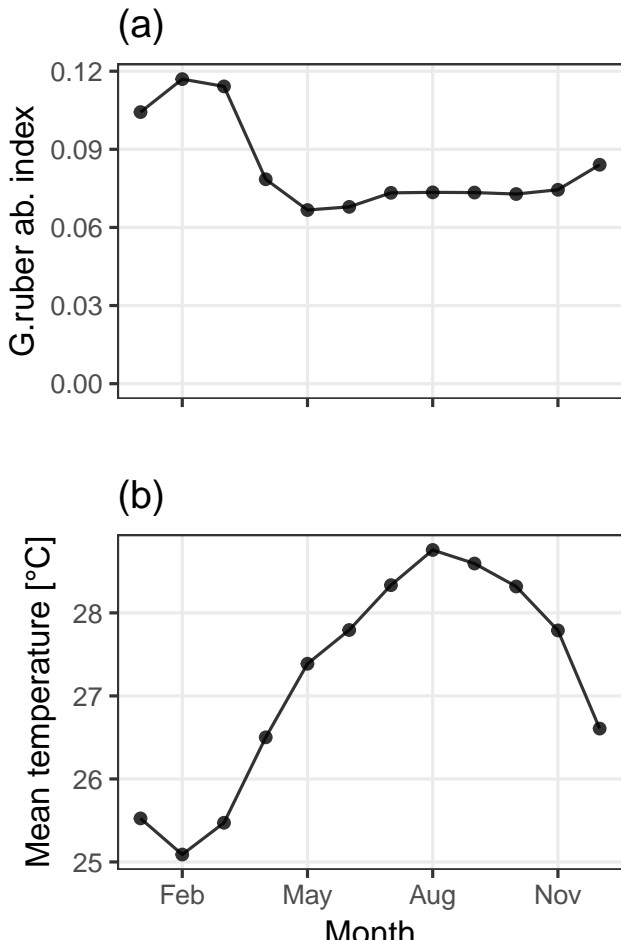

**Figure 2.** Modelled abundance index of *G.ruber* (a), and the mean monthly sea surface temperature (b) at MD97-2141. In this model, *G.ruber* occurs over the whole year with a small maximum during the cooler months of Jan-March, therefore biasing the recorded temperature towards colder temperatures.





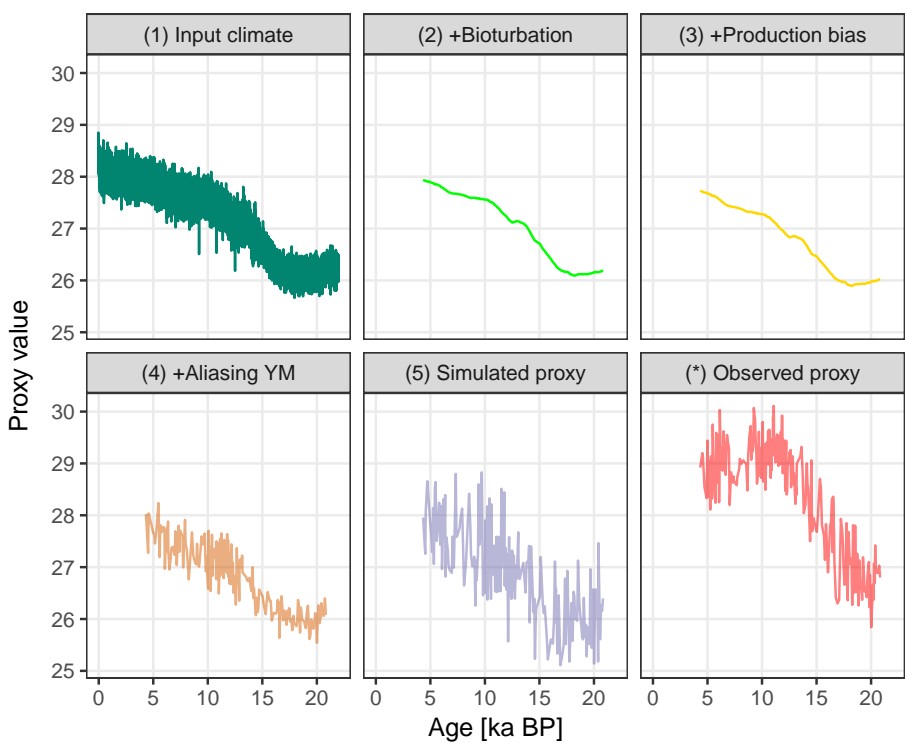

**Figure 3.** A forward modelled foraminiferal Mg/Ca pseudo-proxy record together with the observed Mg/Ca proxy record at core MD97-2141 in the Sulu Sea. The input climate is shown at annual resolution.





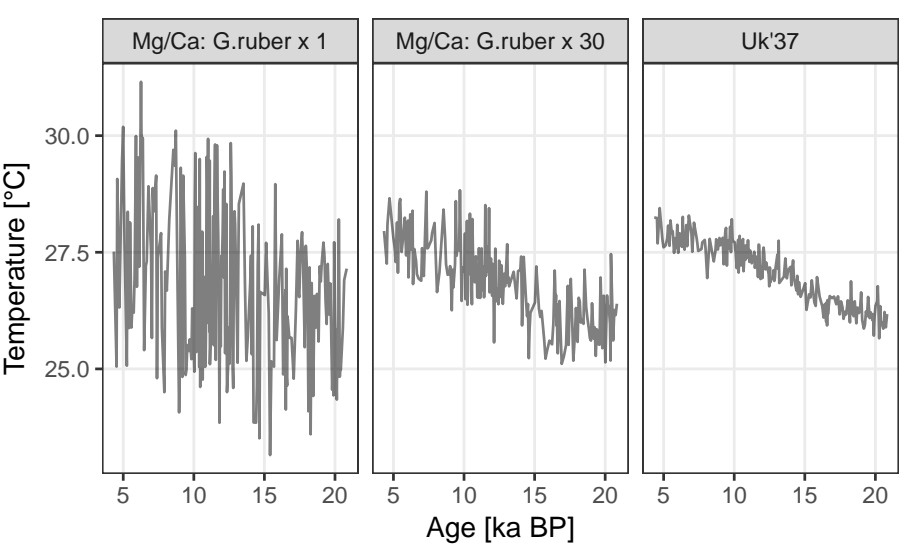

**Figure 4.** Forward modelled proxy based temperature reconstructions for Mg/Ca with 1 and 30 tests of *G.ruber*, and for Uk'37.





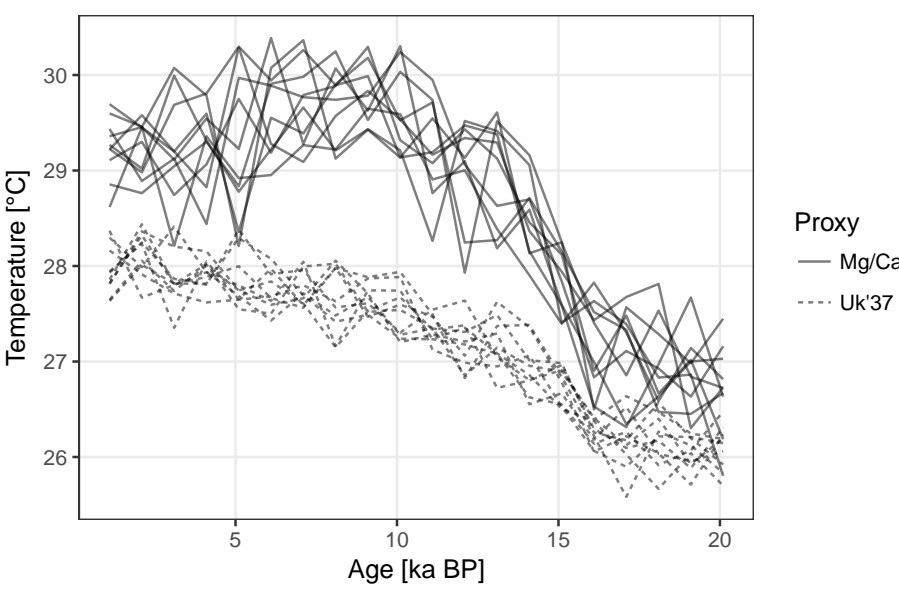

**Figure 5.** Replicate hypothetical Mg/Ca and Uk'37 based records. The two proxy types sample different parts of the seasonal cycle. Ten replicate records are shown for each proxy.





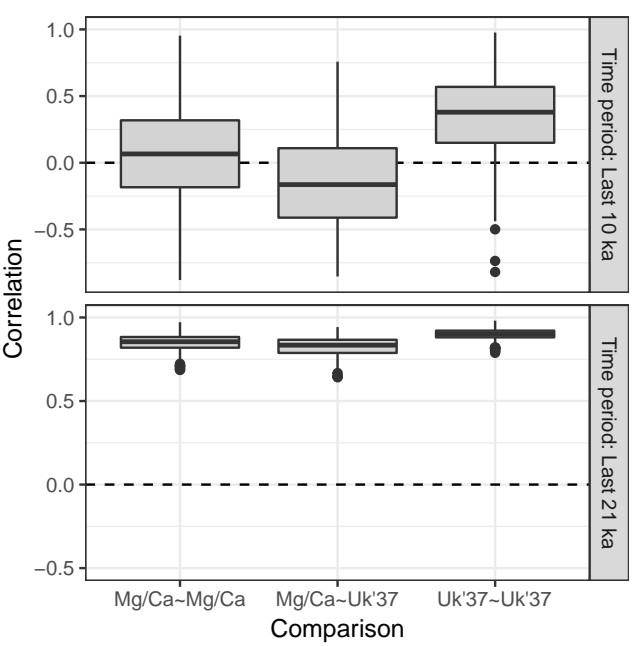

**Figure 6.** Correlation between replicate pairs of forward modelled proxy records.

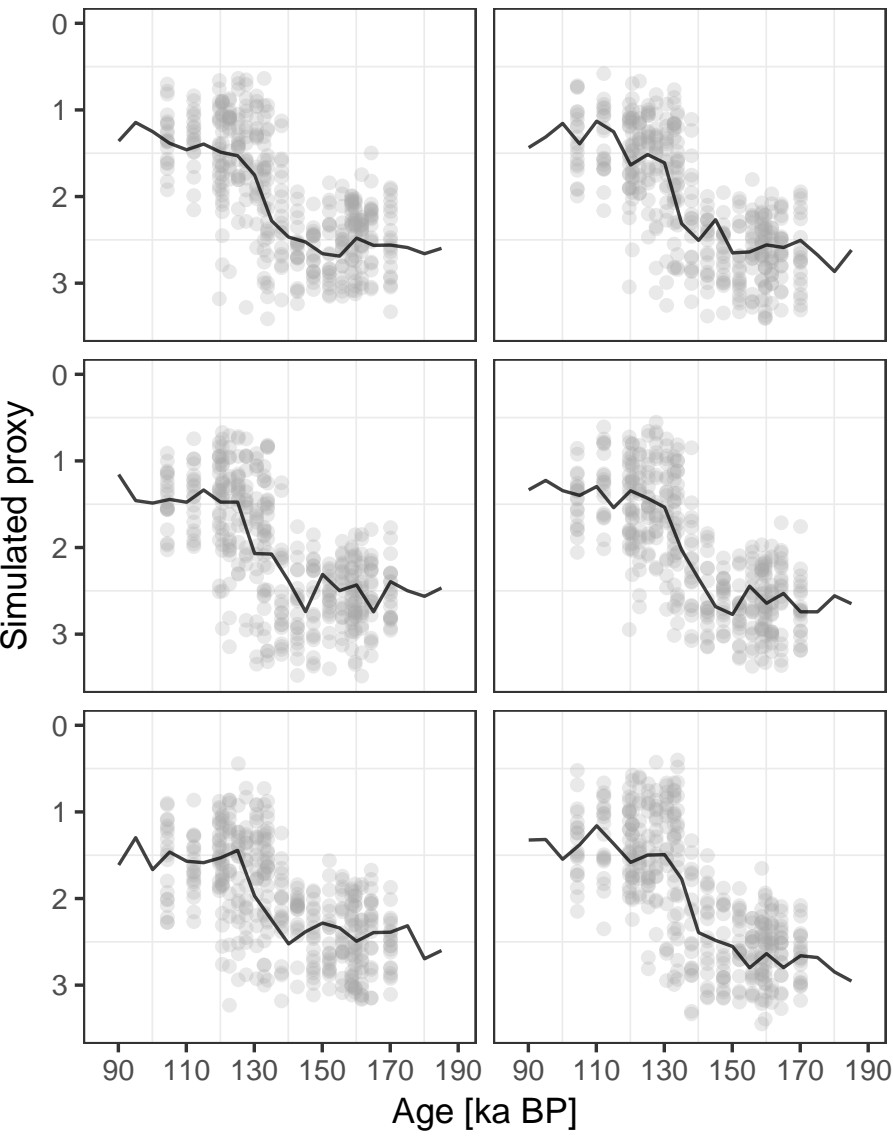

**Figure 7.** Simulated $\delta^{18}$O measured from single foraminiferal tests (circles) and bulk samples (lines). Subplots show six replications with the same parameterisation.





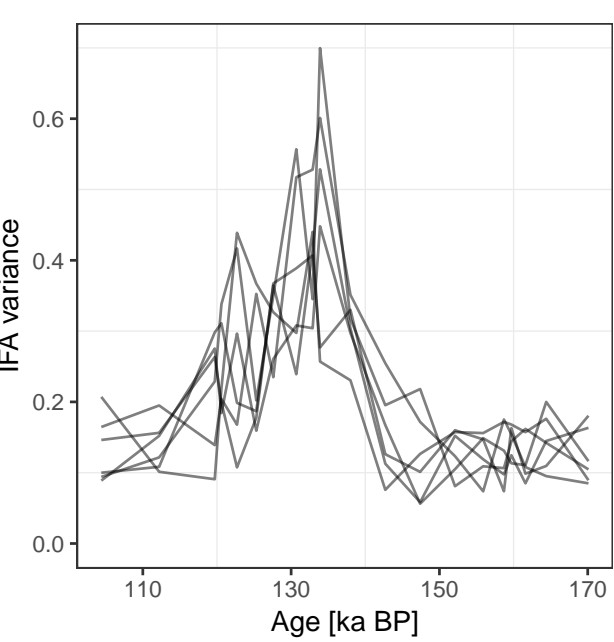

**Figure 8.** Variance in simulated $\delta^{18}O$ measured on sets of 20 individual foraminiferal tests. Lines show six replications with the same parameterisation.





**Table 1.** Required input data and parameters to generate a pseudo-proxy record with *sedproxy*. The final two arguments control the experimental design rather than the proxy record creation process itself.

| Function argument | Description | Possible sources | Default |
|---|---|---|---|
| clim.signal | Input climate signal from which a pseudo-proxy will be forward modelled. | Climate model, instrumental record. | |
| timepoints | Timepoints at which to generate pseudo-proxy values. | Arbitrary, or to match an existing proxy record. | |
| proxy.prod.weights | Proxy production weights provide information on seasonal and habitat (e.g. depth) differences in the amount of proxy material produced. This allows seasonal and habitat biases in the recorded climate to be modelled. | Sediment trap data or dynamic population/biogeochemical model (Fraile et al., 2008; Uitz et al., 2010 ) | all equal |
| bio.depth | Bioturbation depth in cm, the depth down to which the sediment is mixed by burrowing organisms. | Estimated from radiocarbon or from global distribution (Teal et al., 2010). | 10 |
| sed.acc.rate | Sediment accumulation rate in cm ka$^{-1}$. | Sediment core age model. | 50 |
| layer.width | Width of the sediment layer in cm from which samples were taken, e.g. foraminifera were picked or alkenones were extracted. | Core sampling protocol | 1 |
| n.samples | No. of e.g. foraminifera sampled per timepoint. A single number or a vector with one value for each timepoint. | Core sampling protocol | 30 |
| meas.noise | Standard deviation of white noise added to each pseudo-proxy value. | | 0 |
| meas.bias | Each replicate proxy time-series has a constant bias added drawn from a normal distribution with mean = 0, sd = meas.bias. | | 0 |
| n.replicates | Number of replicate pseudo-proxy time-series to simulate from the climate signal. | | |
| smoothed.signal.res | The resolution, in years, of the smoothed (block averaged) version of the input climate signal returned for plotting purposes. If set to NA, no smoothed climate output is generated, this can speed up some simulations. | | 100 |





**Table 2.** Details for sediment core MD97-2141

| Core | Location | Lat | Lon | Proxy | Foram.sp | Reference |
|------|----------|-----|-----|-------|----------|-----------|
| MD97-2141 | Sulu Sea | 8.78 | 121.28 | Mg/Ca | G. ruber | Rosenthal et al., 2003 |