# Peer review of "Sedproxy: a forward model for sediment archived climate proxies"

_Climate of the Past, 2018_

## Referee Comment (RC1) · Anonymous Referee #1 · 27 Mar 2018

Summary:

Dolman and Laepple present a R toolbox (SedProxy) aimed at forward-modeling the various processes contributing to the end signal measured from geochemical measurements on foraminiferal tests or alkenone ratios.

The toolbox can be used to assist in the interpretation of proxy records, perform parameter sensitivity analysis to optimize future study, and to generate pseudo-proxy records to test reconstruction methods.

The authors present four applications for the use of sedproxy, which clearly illustrate the needs to take forward modeling into account in the interpretation of paleoclimate proxy.

[Figure]

As mentioned by the authors, forward modeling is a necessary step toward Bayesian hierarchical modeling and data assimilation. sedproxy provides the foundation for a comprehensive model for sediment archived climate proxies.

Discussion:

Does the paper address relevant scientific questions within the scope of CP? Yes, the manuscript addresses the uncertainty associated with the interpretation of paleoclimate record.

Does the paper present novel concepts, ideas, tools, or data? Yes, the paper presents a new R toolbox for the forward modeling of marine sediment proxies, which is much needed.

Are substantial conclusions reached? Yes, the manuscript clearly illustrate how the use and need of forward modeling using four specific use cases that span the motivation of most paleoclimate studies.

Are the scientific methods and assumptions valid and clearly outlined? For the most part, yes. See specific and minor comments below to clarify some of the assumptions made in this paper.

Are the results sufficient to support the interpretations and conclusions? Yes

Is the description of experiments and calculations sufficiently complete and precise to allow their reproduction by fellow scientists (traceability of results)? Most definitely. I applaud the authors for giving concrete example, with code attached to reproduce their results in the main paper. It also makes comparison of the various functions easier. The applet is also easy to use and allows to compare the effects of tweaking the parameters quickly. The supplementary material contains an .RMD file, which allows to reproduce all the figures contain in the manuscript, while providing enough comments for anyone to follow the results.

Do the authors give proper credit to related work and clearly indicate their own

new/original contribution? For the most part, yes. Few citations are missing. See minor comments below.

Does the title clearly reflect the contents of the paper? Yes.

Does the abstract provide a concise and complete summary? Yes although I would suggest the authors add that sedproxy is an open-source software with open collaboration.

Is the overall presentation well structured and clear? Yes.

Is the language fluent and precise? Yes.

Are mathematical formulae, symbols, abbreviations, and units correctly defined and used? Yes

Should any parts of the paper (text, formulae, figures, tables) be clarified, reduced, combined, or eliminated? No.

Is the amount and quality of supplementary material appropriate? Most definitely. Contains the package and a .RMD file to reproduce the results presented in this paper. This file can also be used as a template for anyone who wishes to plot their own results from sedproxy.

Specific Comments:

The assumptions that sedproxy makes are presented in the last section of the manuscript. I would suggest moving them upfront to help the reader follow along.

The mathematical formulation of the transformation from Mg/Ca (and UK'37) to temperature is not clear in the text. Which calibration is being used? Can the user input one of their choice? On lines 13-14 (page 3), the authors talk about secondary influences on these proxies but they don't seem to be take into account in the forward model. One of the advantages of forward modeling is to be able to take into consideration more complex calibration equations. Why not do this here?

Minor Comments:

The introduction is often lacking in proper citations. For instance, it's missing a citation on page 1, line 22 about the use of Mg/Ca as a paleothermometer or examples of downcore records.

Page 4, lines 10-11: dissolution effects may not be minimal and may be missed during cleaning/processing if SEM images were not taken. See the manuscript by Hertzberg and Schmidt, 2013, EPSL (doi: 10.1016/j.epsl.2013.09.0444). The authors should reword this comment and add this assumption to their list of assumptions and caveats.

Move the discussion about INFAUNAL from section 8 to section 7.

---

## Referee Comment (RC2) · B. Metcalfe (Referee) · 16 Apr 2018

B. Metcalfe

Quantifying uncertainty within palaeoclimate archives is essential if we are to make better predictions of the past, in light of this Dolman and Laepple (2018) have produced a R package, sedproxy, that combines the seasonal weighting of Mix (1987), the mixing through bioturbation of Berger and Heath (1968) and the measurement noise within Laepple and Huybers (2013) that is handy and easy to use. The authors have provided examples on a couple of applications that showcase the potential of their code. These examples are logical for the most part, however why not continue this trend and provide an example of IFA based upon MD97-2141 by expanding Figure 4. Moving

the discussion of Scussolini et al's (2013) results til later, that way a new-user has an example with the same dataset from start to finish?

The problem with this paper though, is that whilst it is needed by the community the authors seem to be presenting code that is more a version 0.5 as outlined, throughout the text, by the authors themselves. Throughout the text the authors offer suggestions of 'easy' improvements that they could do to their own code, which is commendable. However, in a couple of instances they note that other code, by other groups, exists that does a similar job and in some parts this weakens the whole. For instance, sedproxy's "season and depth habitat in the recording (but not necessarily the climate) is assumed to be invariant over time" which contradicts Mix's temperature based weighting of the seasonal signal (Roche et al., 2017). The authors state that this will help to compare models with proxy data, but if the monthly weighting is static through time can't someone bypass sedproxy and compare model-March, or a seasonal weighted, output with G. ruber Mg/Ca directly? Likewise, is March really equivalent through time?

The model also uses the same units as the input series, "we do not explicitly model the encoding process for specific sensors. Other tools have been developed to do this….and could be used to pre-process the input climate signal" with the authors suggesting that "a back-transformation can then be applied to the generated pseudo-proxy records, which itself might model uncertainty by varying the parameters of the calibration". My question, why not cut out the middle man in which they risk being supplanted by the code of others and add this into their code? In trace metal geochemistry the calibration(s) of Mg/Ca vs. Temperature is by far one source of error that is overlooked repeatedly. Likewise, the authors should consider who will be their end-user (e.g., whether some end-users may or may not be comfortable with or take the time with pre-processing the data using other code). Therefore, I think the paper could benefit greatly from expansion of the code in ways that the authors themselves list.

Specific comments

(Pg. 3 Line 11-12) "we do not explicity model the encoding process for specific sensors" maybe explicity state for clarity that sedproxy doesn't model conversion between temperature and Mg/Ca or Uk37, i.e. calibrations are not used. As it is not clear, as demonstrated by Reviewer 1: " The mathematical formulation of the transformation from Mg/Ca (and UK'37) to temperature is not clear in the text. Which calibration is being used? Can the user input one of their choice?" Perhaps making this clear earlier (on page 3) like you do later at pg 14 line 31 – pg 15 line 5 would benefit the readership.

(Pg. 4, Line 10) "We assume here that these effects are minimal" Dissolution is far from minimal, the lysocline is a marked boundary because it is when dissolution becomes apparent (because the rate of dissolution increases) but dissolution is still occurring above the lysocline. Berger suggested that only a small percentage of the flux reaches the seafloor / ends up preserved. If one were to consider it theoretically, productive months (rich in Corg) will likely lead to increased benthic activity and increased CaCO3 dissolution. The authors acknowledge that sedproxy doesn't include a flux component (pg. 15 lines 6-16), if they do add in such a component, it is worth considering that some seafloor processes might also be seasonally driven (or driven by seasonal flux of food/organic matter that can be respired, to the seafloor).

(Pg. 4 Line 16) "Due to bioturbation these individuals will be a mixed sample that integrate the climate signal over an extended time period" I would disagree that this is solely a function of bioturbation, low sedimentation rate (e.g. 1 cm per kyr) means that individuals are from potentially any point within 1000 years irrespective of benthic seafloor processes. Perhaps mention here, that low SAR is already a 'smoothed'-integrated record regardless of bioturbation.

(Pg. 7 Section 3.3) It would benefit the reader, and add clarity, if the authors better express this section so that sedproxy doesn't become a black box. The independent error term for each proxy type, am I correct in assuming that this is the same as: (Laepple and Huybers 2013; Section 5. Application of the correction filter) "each record requires estimating the two adjustable parameters that define the background variability: the

spectral slope (beta) and the standard deviation associated with (eta). We perform an exhaustive search over the values of beta = (0, 0.1,...1.9, 2.0) and STD(eta) = (0, 0.05,...1.95, 2.0), searching for the pair of values that minimize the mean square deviation between the logarithm of the observed sepectra and logarithm of the model spectra." later on in the same 2013 paper stating "and a 0.25 and 0.45 standard deviation of $\eta$ is prescribed for Uk37 and Mg/Ca respectively". I think, within the text of this paper, the authors need to justify the value of the standard deviation of their gaussian random variable, how it is constructed for each proxy, its limitation etc. As this will essentially create a model-specific result.

Furthermore, how is cleaning of foraminiferal tests parameterized? Would using the Mg/Ca variability in culture studies be a better source of inter-individual variability parameterization (i.e. one question to ask is, is inter-individual variability constant over values of temperature cultured or does it vary)?

(Pg. 10, Line 1&2) "the input climate signal smoothed to centennial resolution" why have the authors smoothed the input variable? Does this not contradict the point of the model? Furthermore, how was it smoothed, which method? It is only mentioned here and table 1 (where "block average" smoothing is identified) that there is mention of smoothing in the record, this should be stated within the main text.

Figure 3 – would it not be better in panel one (input climate) to show the annual minimum or maximum (as a shading)? Your model has a seasonal weighting component therefore the 'full range' should be included, at present the figure at a glance (without reading the caption) appears to show a narrow temperature window. It also makes it difficult to envision the seasonal weighting. Furthermore, might it be prudent to show the measured proxy values of temperature in more than one panel (other than panel 6)? At least plot the forward model and proxy result together in panel 5. Additonally, what is the error on the reconstructed temperature from Mg/Ca?

(Pg. 11 Section: Influence of the number of foraminifera per sample) Is figure 4 only a

single run of each n = 1 and n = 30 for G. ruber? If so, would it not be better to produce a figure similar to Figure 5 with replicates. It would/might show that replicates of n = 1 have a larger spread than replicates of n = 30... or not.

(Pg. 13 Section 7) Globorotalia truncatulinoides is a deep dwelling planktonic foraminifera (∼500 m), the rationale behind Scussolini et al.'s species selection was that deeper dwellers would exhibit perturbations within the water mass through the movement of Aghulus leakage rings. Therefore, what is the rationale for adding a seasonal component (Pg. 13 Line 14) in waters >500 m that have little seasonality? (see figure 2 in Scussolini and Peeters 2013)

Also and this is just a point of note regarding Figure 7's mean of 45 foraminifera: larger planktonic varieties (such G. truncatulinoides) are generally heavy, most modern mass spectrometers have an upper or lower end in weight, the standard number of foraminifera that constitute 'bulk samples' of heavy foraminifera is 3-5 specimens (i.e. Cleroux et al., 2013 used 10-25 specimens to make four aliquots, x2 for trace metal and x2 for stable isotope geochemistry). Scussolini and Peeters 2013 took a small portion of a large number of shells thus negating this weight limit: "Between 35 and 55 shells for each species were crushed, and a portion of approximately 150 '$\mu$g of homogenized calcite fragments was used for stable isotope analysis. This approach was adopted to maximize the number of shells involved and therefore the analyses' representativeness of the foraminiferal population". In the past measurements came from samples with more specimens, that is not the case today, so perhaps a mean with fewer specimens would be more fitting?

(Pg. 14 Line 23) "this enables more quantitative comparisons to be made between climate models and proxy data than would classical direct comparison" whilst sedproxy is for the most part better (theoretically) than a simple comparison of proxy data with Mean annual temperature provided by models, does the fact that neither season or depth vary add its own source of error?

(Pg. 15 Line's 32-35) The funnel effect, at least in sediment traps, in which foraminifera deposited may in fact be 'expatriates' does certainly suggest that foraminifera may not have a signal that is directly related to that above the core site. Personally, however if you combine the depth integrated growth (e.g. Wilke et al. 2006 and references therein) with the suggestion in culture of precipitation of calcite on preceding chambers then for the most part the signal preserved within a shell will be overprinted by the final chamber's signal, or a depth weighted function (Roche et al., 2017). Therefore, a model would need only to take into account the distance covered following mortality (settling speed ~1-2 days from surface to sediment)

Technical

Pg. 1, Line 21: Remove 'marine', replace with planktonic or pelagic Pg. 2, Line 13: Would Mix 1987 and/or Mulitza et al., 1997 not be more appropriate references for 'the influence of seasonal recording' Pg. 3, Line 21: remove duplicate 'thus' Pg. 5 line 3: change 'or' to 'including', as vital effects (the potential metabolic effects) are not exclusively inter-individual variation (given the individual life histories of foraminifera found within the sediment and or plankton tow samples Pg. 13 Line 9 'choose parameter values resembling this study' but then state further 'these choices are partly arbitrary' Pg. 15 line's 25-27 The scenario envisioned is performed by Lougheed et al. (2017)

References Scussolini, P. and Peeters, F. J. C.: A record of the 460 thousand years of upper ocean stratification from the central Walvis Ridge, South Atlantic, Paleoceanography, 28, 426–439, doi: 10.1002/palo.20041, 2013 Lougheed, B., Metcalfe, B., Ninnemann, U., and Wacker, L., Moving beyond the age-depth model paradigm in deep sea palaeoclimate archives: dual radiocarbon and stable isotope analysis on single foraminifera, Clim. Past Discussions. 2017 Wilke, Bickert, Peeters, The influence of seawater carbonate ion concentration [CO32-] on the stable carbon isotope composition of the planktic foraminifera species Globorotalia inflata, Marine Micropalaeontology, 58, 243-258, 2006

---

## Short Comment (SC1) · 16 Apr 2018

The authors utilize our study (Scussolini et al., 2013) to illustrate one of the applications of their new model for marine sediment proxies. With the aim of suggesting improvements to this interesting manuscript, I bring to attention some relevant aspects of Dolman and Laepple's handling of that study.

First, Scussolini et al. (2013) analysed the planktic foraminifer Globorotalia truncatulinoides (sinistral coiling variety). This organism calcifies at depths beyond 400 or 600 m, according to the relevant literature and to Scussolini and Peeters (2013, Paleoceanography; doi: 10.1002/palo.20041; see also references therein), who compared values form core-top specimens to modern hydrography. At these depths, at the core

site, there is hardly any seasonal variation in temperature and salinity. To assume 0.5 ‰ seasonal noise to mimic the del18O signal seems therefore inappropriate. I expect that this shouldn't change the position or magnitude of the peak in variability simulated by 'sedproxy', but it would be advisable to rectify the calculations to reflect this.

Second, Scussolini et al. (2013) report that they 'corrected the variance of foraminiferal del18O by subtracting that of external calcite standards measured in the same sequence', with the aspiration to clean their proxy from the spurious effect of measurement noise. It seems that Dolman and Laepple do not take this into account, as they 'assume a measurement noise of 0.1‰ del18O for the IFA and the bulk measurements'.

Third, Dolman and Laepple assumed 'a climate transition from 0.4 ‰ at 190 ka BP, to 2.6 ‰ at 90 ka BP'. The signal in core 64PE-174P13 goes from ca. 1.6 ‰ at 190 ka BP, to 1.3 ‰ at 90 ka BP (see fig. 2 in Scussolini et al. 2013). Where were the values of 0.4 and 2.6 ‰ taken from? In any case, this choice of such extended time frame is puzzling, as the sharp change in del18O occurs obviously across the glacial termination (ca. 140 to 125 ka BP). I would recommend that the authors take a more meaningful time frame. Also, they may consider using a more realistic representation of the transition in sea water isotopic values than the logistic function (taking inspiration from global del18O stacks, or a sea-level reconstruction?).

Further, assuming bioturbation reaching 10 cm from the top of the sediment will obviously produce a peak in variability in any record across a signal transition such as a glacial termination. While it is unrealistic to think that bioturbation is absent from core 64PE-174P13, Scussolini et al. (2013) advanced multiple lines of reasoning to exclude strong bioturbation in core 64PE-174P13, not least visible laminations in parts of the record (see also the author's response to referee #1, who raised specifically the point of bioturbation: https://www.clim-past-discuss.net/9/C511/2013/cpd-9-C511-2013.pdf). An additional argument against the role of bioturbation and in favor of an interpretation of the variability signal as proxy for Agulhas rings comes from Scussolini et al. (2015,

Geology, doi: 10.1130/G36238.1). There, a tight coupling is shown between the Agulhas rings proxy with the ice-volume-corrected seawater del18O of G. truncatulinoides, a proxy for the high salinity anomalies that Agulhas rings seem to have introduced at the core location (see below a snapshot of the relevant figure in Scussolini et al. 2015, showing the two proxies). It is important to note that the two proxies are analytically independent of one another. It is not clear from the manuscript whether the authors have reasons to prefer the interpretation of the signal in terms of bioturbation.

As the authors admit, the choice of parameters here was 'partly arbitrary.' I explained four ways in which these choices seem needlessly arbitrary and indeed inadequate, and I suggested ways to improve these choices. One interesting application of 'sed-proxy' may in fact be that of revealing the bioturbation depth that would be sufficient to explain the variability peak shown for core 64PE-174P13, if the hypothesis of its ring-origin were to be rejected.

[Figure]

**Fig. 1.**

---

## Author Comment (AC1) · 25 Jun 2018

Dear reviewer,

Thank you for taking the time to review our discussion paper. We thank you for the constructive comments and plan to make several important changes in response to your suggestions. We respond to your comments below and have included relevant portions of the review below (in blue italicised text).

*"Do the authors give proper credit to related work and clearly indicate their own new/original contribution? For the most part, yes. Few citations are missing."*

We will check thoroughly for missing citations and add these to the revised version.

*"I would suggest the authors add [to the abstract] that sedproxy is an open-source software with open collaboration."*

We will modify the abstract to make clear that sedproxy is open source and that contributions are welcome.

**Specific Comments:**

*"The assumptions that sedproxy makes are presented in the last section of the manuscript. I would suggest moving them upfront to help the reader follow along."*

We will list the assumptions earlier in the "implementation" section (section 3).

*"The mathematical formulation of the transformation from Mg/Ca (and UK'37) to temperature is not clear in the text. Which calibration is being used? Can the user input one of their choice?"*

In the version of sedproxy presented in the discussion paper we did not in fact deal at all with calibration or its uncertainty. We have now modified the code to allow an input climate signal to be converted from temperature to proxy units using either the Uk'37 calibration from Müller et al (1998), or (one of) the Mg/Ca to temperature calibrations from Anand et al (2003). Alternatively, the user can supply their own parameter values for the calibration slope and intercept or pre-convert the input climate signal. Uncertainty in the calibration can be examined by applying the calibration using parameters drawn from a bivariate distribution representing the uncertainty in the fitted slope and intercept parameters. We will update the manuscript and package documentation to give examples of this.

*"On lines 13-14 (page 3), the authors talk about secondary influences on these proxies but they don't seem to be take into account in the forward model. One of the advantages of forward modelling is to be able to take into consideration more complex*

*calibration equations. Why not do this here?"*

These kinds of secondary effect could be included with a user-supplied calibration function, but we think it is beyond the scope of this paper to actually suggest more advanced calibrations.

**Minor Comments:**

*"The introduction is often lacking in proper citations. For instance, it's missing a citation on page 1, line 22 about the use of Mg/Ca as a paleo-thermometer or examples of down-core records."*

We agree that it would be useful to the reader if more background references are provided and will do a thorough revision of the citations in the manuscript.

*"Page 4, lines 10-11: dissolution effects may not be minimal and may be missed during cleaning/processing if SEM images were not taken. See the manuscript by Hertzberg and Schmidt, 2013, EPSL (doi: 10.1016/j.epsl.2013.09.0444). The authors should reword this comment and add this assumption to their list of assumptions and caveats."*

We will expand our discussion of the possible effects of dissolution and add this to the assumptions, which will also be listed earlier in the manuscript.

*"Move the discussion about INFAUNAL from section 8 to section 7."*

We will discuss INFAUNAL in a revised version of section 7.

Once again we thank you for your comments,

Andrew Dolman.

**References.**

Anand, P., Elderfield, H. and Conte, M. H.: Calibration of Mg/Ca thermometry in planktonic foraminifera from a sediment trap time series, Paleoceanography, 18(2), 1050, doi:10.1029/2002PA000846, 2003.

Müller, P. J., Kirst, G., Ruhland, G., von Storch, I. and Rosell-Melé, A.: Calibration of the alkenone paleotemperature index U37K' based on core-tops from the eastern South Atlantic and the global ocean (60 N-60 S), Geochim. Cosmochim. Acta, 62(10), 1757–1772, doi:10.1016/S0016-7037(98)00097-0, 1998.
* * *

---

## Author Comment (AC2) · 25 Jun 2018

Dear Brett Metcalfe,

We appreciate your taking the time to review our discussion paper and thank you for your constructive and detailed comments that will help to improve the manuscript. We first respond to the main points in your general discussion of the paper and then to the specific comments. We have included portions of your review as blue italicised text.

**Response to general discussion.**

*"The problem with this paper though, is that whilst it is needed by the community the*

[Figure]

*authors seem to be presenting code that is more a version 0.5 as outlined, throughout the text, by the authors themselves. Throughout the text the authors offer suggestions of 'easy' improvements that they could do to their own code, which is commendable. However, in a couple of instances they note that other code, by other groups, exists that does a similar job and in some parts this weakens the whole."*

We do not fully agree with the general characterisation of the sedproxy package as "version 0.5 code". While the processes have been described by other groups (or in previous publications from our group) in nearly all instances, no user-friendly code existed that implements these processes. This is clearly visible in the literature that largely continues to ignore most of the effects. As we describe below, we have added functionality to address your specific points about variable versus static habitat weights and to address proxy calibration uncertainty.

**Habitat weightings.**

*"The authors state that this will help to compare models with proxy data, but if the monthly weighting is static through time can't some-one bypass sedproxy and compare model-March, or a seasonal weighted, output with G. ruber Mg/Ca directly? Likewise, is March really equivalent through time?"*

While sedproxy could be bypassed – and a single month or seasonally weighted average from a climate model compared directly to a proxy record – such a comparison would ignore the effects of bioturbation, seasonal biases, aliasing of seasonal and inter-annual variability and measurement error. Therefore, sedproxy which includes these first order effects is a useful tool even with the limitation of static habitat weights, and strongly expands on the classical direct interpretation. However, we also see that the current practise of assuming no seasonality or fixed seasons (e.g. Leduc et al. 2013, Lohmann et al., 2013, Marcott et al. 2013, Shakun et al. 2012) is not optimal. We have therefore extended the model to allow for non-static seasonal-habitat or depth-habitat

weights. We have modified the code so that a matrix of weights of the same size as the input climate matrix can be passed in place of a vector of static weights, or a named function plus arguments that will return weights as a function of the input climate. In this way, non-static season/habitat weights can be pre-calculated using either the simple Gaussian response approach of Mix (1987), or something more advanced such as the proposed FAME module (Roche et al 2017). We have ported the relevant functions and data objects from FAME v1.0 Python module (Roche et al 2017) and will include them in the sedproxy R package (under the appropriate GPL license). The calculation of weights from the input climate can be done either within R, or externally with whatever model the user prefers.

Applied to Example 1 from our manuscript, using dynamic habitat weighting from the FAME parametrisation results in an apparent mean temperature change between the glacial (18 ka BP) and the mid-Holocene (5 ka BP) of 1.63 °C, compared to 1.75 °C using static weights derived using PLAFOM with modern day conditions (see Fig. 1). In this example, the difference between static and dynamic weights is small but still illustrates the potential for adaptive behaviour of proxy signal carriers to lead to an underestimation of the magnitude of climate shifts. This effect could be larger for a record from a region with a larger seasonal cycle and/or taxon with a more pronounced seasonality in its productivity. We will expand one of the examples to illustrate the use of dynamic habitat weighting.

**Calibration**

*"The model also uses the same units as the input series, "'we do not explicitly model the encoding process for specific sensors. Other tools have been developed to do this ... and could be used to pre-process the input climate signal'" with the authors suggesting that "'a back-transformation can then be applied to the generated pseudo-proxy records, which itself might model uncertainty by varying the parameters of the*

*calibration'". My question, why not cut out the middle man in which they risk being supplanted by the code of others and add this into their code? In trace metal geochemistry the calibration(s) of Mg/Ca vs. Temperature is by far one source of error that is overlooked repeatedly. Likewise, the authors should consider who will be their end-user (e.g., whether some end-users may or may not be comfortable with or take the time with pre-processing the data using other code). Therefore, I think the paper could benefit greatly from expansion of the code in ways that the authors themselves list."*

We have modified the sedproxy code to add several options for modelling calibration uncertainty. If the argument "proxy.calibration.type" is set to either 'UK37' or 'MgCa', the input climate matrix will be converted using the Uk'37 to temperature calibration from Müller et al (1998), or (one of) the Mg/Ca to temperature calibrations from Anand et al (2003). Alternatively, the input climate matrix and measurement errors can be pre-transformed by the user, the "proxy.calibration.type" is then left at its default value of 'identity'.

Uncertainty in the relationship between temperature and proxy units can be examined by requesting multiple replicate pseudo-proxies. In this case, for each replicate a random set of calibration parameters are drawn from a bivariate normal distribution that represents the uncertainty in the fitted calibration model. The bivariate distributions are parametrised by mean values for the regression coefficients, plus their variance covariance matrices. We have estimated these for the supplied calibrations by refitting regression models to the calibration data used in the original publications (details will be given in a supplement).

Both the Mg/Ca and Uk'37 calibration functions will also accept optional arguments that replace their default parameter values and variance-covariance matrices. For alternative calibration models that have a different functional form, (for these or other proxy types), the name of a user supplied function can be passed that will do the calibration conversion. A template for a user defined function will be given in the documentation.

We have also modified the default plotting functions so that the additional calibration uncertainty is shown.

**Response to specific comments:**

*"(Pg. 3 Line 11-12) "we do not explicitly model the encoding process for specific sensors" maybe explicitly state for clarity that sedproxy doesn't model conversion between temperature and Mg/Ca or Uk37, i.e. calibrations are not used. As it is not clear, as demonstrated by Reviewer 1: "The mathematical formulation of the transformation from Mg/Ca (and UK'37) to temperature is not clear in the text. Which calibration is being used? Can the user input one of their choice?" Perhaps making this clear earlier (on page 3) like you do later at pg 14 line 31 – pg 15 line 5 would benefit the readership."*

We have added explicit conversion to and from proxy units, including a method to model uncertainty in this conversion (see above). We will modify the manuscript and documentation accordingly.

*"(Pg. 4, Line 10) "We assume here that these effects are minimal" Dissolution is far from minimal, the lysocline is a marked boundary because it is when dissolution becomes apparent (because the rate of dissolution increases) but dissolution is still occurring above the lysocline. Berger suggested that only a small percentage of the flux reaches the seafloor / ends up preserved. If one were to consider it theoretically, productive months (rich in Corg) will likely lead to increased benthic activity and increased CaCO3 dissolution. The authors acknowledge that sedproxy doesn't include a flux component (pg. 15 lines 6-16), if they do add in such a component, it is worth considering that some seafloor processes might also be seasonally driven (or driven by seasonal flux of food/organic matter that can be respired, to the seafloor)."*

We will expand the discussion of dissolution in the manuscript text and highlight that this is not modelled and that it may itself have a seasonally driven component.

*"(Pg. 4 Line 16) "Due to bioturbation these individuals will be a mixed sample that integrate the climate signal over an extended time period" I would disagree that this is solely a function of bioturbation, low sedimentation rate (e.g. 1 cm per kyr) means that individuals are from potentially any point within 1000 years irrespective of benthic seafloor processes. Perhaps mention here, that low SAR is already a 'smoothed'-integrated record regardless of bioturbation."*

The function to calculate bioturbation weights does take into account the width of the sediment layer from which a sample of forams is picked, or Uk'37 extracted (argument "layer.width"). Specifically, it is a convolution with a uniform probability density function (PDF) and the exponential PDF generated by the bioturbation. We will make this detail clearer in the text.

*"(Pg. 7 Section 3.3) It would benefit the reader, and add clarity, if the authors better express this section so that sedproxy doesn't become a black box. The independent error term for each proxy type, am I correct in assuming that this is the same as: (Laepple and Huybers 2013; Section 5. Application of the correction filter) "each record requires estimating the two adjustable parameters that define the background variability: the spectral slope (beta) and the standard deviation associated with (eta). We perform an exhaustive search over the values of beta = (0, 0.1,...1.9, 2.0) and STD(eta) = (0, 0.05,...1.95, 2.0), searching for the pair of values that minimize the mean square deviation between the logarithm of the observed spectra and logarithm of the model spectra." later on in the same 2013 paper stating "and a 0.25 and 0.45 standard deviation of eta is prescribed for Uk37 and Mg/Ca respectively". I think, within the text of this paper, the authors need to justify the value of the standard deviation of their Gaussian random variable, how it is constructed for each proxy, its limitation etc. As this will essentially create a model-specific result."*

We agree that we should have been clearer about the parametrisation we used for the independent error term. The value of the independent error term is something that the

user should decide and justify for a given study. However, as we give suggested values in the manuscript and documentation it is likely that these will be used as "defaults".

It was apparent from the work in Laepple and Huybers 2013 that even after accounting for aliasing and measurement error, there was additional unaccounted independent error in Mg/Ca and Uk'37 proxy records. The magnitude of this error was estimated by tuning a noise parameter to obtain the best fit between power spectra for proxy and pseudo-proxy records. Further, it was shown that these empirically derived parameters were consistent with independent estimates from replicate measurements of Mg/Ca and Uk37. Most datasets contributing to Laepple and Huybers 2013 were based on a similar number of foram tests. Thus a single parameter was a valid approximation even if parts of the true error are to a first order independent of the number of foraminiferal tests (e.g. analytical error) whereas other errors (such as the habitat depth range that was not accounted for in Laepple and Huybers 2013) scale with the sample size.

As sedproxy should be applicable independent of the number of foram tests per sample, we propose to split the independent error term into 2 parts, *sigma.measurement* and *sigma.individual*. *sigma.measurement* will encompass both the analytical error of the measurement process and any other sources of error that are introduced during the preparation of the sample (e.g. cleaning for Mg/Ca). *sigma.individual* will describe all remaining variations, for example inter-individual variations or the depth habitat if unaccounted for. This error will scale with the number of individuals and is likely to be site and species dependent, although the empirical estimates of the sum of both error terms in Laepple and Huybers suggested similar values between study sites. We will describe both error terms and the proposed default values in detail in the revised manuscript.

*"(Pg. 10, Line 12) "the input climate signal smoothed to centennial resolution" why have the authors smoothed the input variable? Does this not contradict the point of the model? Furthermore, how was it smoothed, which method? It is only mentioned*

*here and table 1 (where "block average" smoothing is identified) that there is mention of smoothing in the record, this should be stated within the main text."*

The smoothing is only for display purposes as the annual or monthly variance of the input climate signal is typically so much larger than the processed pseudo-proxy (or real proxy reconstruction) that the plots become unreadable. The forward model always works with the full resolution of the input. This is stated in table 1 and we will make it clearer in the figure legends.

*"Figure 3 – would it not be better in panel one (input climate) to show the annual minimum or maximum (as a shading)? Your model has a seasonal weighting component therefore the 'full range' should be included, at present the figure at a glance (without reading the caption) appears to show a narrow temperature window. It also makes it difficult to envision the seasonal weighting. Furthermore, might it be prudent to show the measured proxy values of temperature in more than one panel (other than panel 6)? At least plot the forward model and proxy result together in panel 5. Additionally, what is the error on the reconstructed temperature from Mg/Ca?"*

We will add the monthly resolution climate information behind the smoothed version in figure 3 panel 1. We will also combine the bioturbated signal and habitat biased signal in one panel (currently panels 2-3) so that the bias is easier to judge. Similarly, the simulated and observed proxy records will be shown together. This will free-up space to also show the calibration uncertainty in an additional panel.

*"(Pg. 11 Section: Influence of the number of foraminifera per sample) Is figure 4 only a single run of each n = 1 and n = 30 for G. ruber? If so, would it not be better to produce a figure similar to Figure 5 with replicates. It would/might show that replicates of n = 1 have a larger spread than replicates of n = 30. . . or not."*

We like this idea and are testing how best to include this. A candidate figure is included here (Fig. 2).

*"(Pg. 13 Section 7) Globorotalia truncatulinoides is a deep dwelling planktonic foraminifera (~500 m), the rationale behind Scussolini et al.'s species selection was that deeper dwellers would exhibit perturbations within the water mass through the movement of Aghulus leakage rings. Therefore, what is the rationale for adding a seasonal component (Pg. 13 Line 14) in waters >500 m that have little seasonality? (see figure 2 in Scussolini and Peeters 2013)"*

In response to the comments from Paolo Scussolini (SC1) we will be adjusting the parameters used for this example to be more realistic. This includes a much-reduced seasonal cycle.

*"Also and this is just a point of note regarding Figure 7's mean of 45 foraminifera: larger planktonic varieties (such G. truncatulinoides) are generally heavy, most modern mass spectrometers have an upper or lower end in weight, the standard number of foraminifera that constitute 'bulk samples' of heavy foraminifera is 3-5 specimens (i.e. Cleroux et al., 2013 used 10-25 specimens to make four aliquots, x2 for trace metal and x2 for stable isotope geochemistry). Scussolini and Peeters 2013 took a small portion of a large number of shells thus negating this weight limit: "Between 35 and 55 shells for each species were crushed, and a portion of approximately 150 $\mu$g of homogenized calcite fragments was used for stable isotope analysis. This approach was adopted to maximize the number of shells involved and therefore the analyses' representativeness of the foraminiferal population". In the past measurements came from samples with more specimens, that is not the case today, so perhaps a mean with fewer specimens would be more fitting?"*

We agree that today, many measurements are performed with fewer specimens. However Figure 7 specifically deals with Scussolini et al. 2013. For this study, as long at the sample was well homogenized, using a mean of 45 foraminifera should be the best approximation to their procedure as the Mg/Ca signal from 35-55 individuals should be present in each of the "bulk" data points in their figure 2.

*"(Pg. 14 Line 23) "this enables more quantitative comparisons to be made between climate models and proxy data than would classical direct comparison" whilst sedproxy is for the most part better (theoretically) than a simple comparison of proxy data with Mean annual temperature provided by models, does the fact that neither season or depth vary add its own source of error?"*

We have updated sedproxy so that varying habitat weights can be used. Not having varying seasonal and depth habitats does not add a source of error, rather it leaves in a source of error that would still be there with a simple model-data comparison.

*"(Pg. 15 Line's 32-35) The funnel effect, at least in sediment traps, in which foraminifera deposited may in fact be 'expatriates' does certainly suggest that foraminifera may not have a signal that is directly related to that above the core site. Personally, however if you combine the depth integrated growth (e.g. Wilke et al. 2006 and references therein) with the suggestion in culture of precipitation of calcite on preceding chambers then for the most part the signal preserved within a shell will be overprinted by the final chamber's signal, or a depth weighted function (Roche et al., 2017). Therefore, a model would need only to take into account the distance covered following mortality (settling speed ~1-2 days from surface to sediment)"*

It is reassuring to know that sedimented forams provide a relatively local signal, however we also deal here with organic proxies which have much greater potential for lateral transport (e.g. Mollenhauer et al. 2003). As this is a general discussion, we would like to keep this qualification.

**Technical comments**

*Pg. 1, Line 21: Remove 'marine', replace with planktonic or pelagic*

Agreed.

*Pg. 2, Line 13: Would Mix 1987 and/or Mulitza et al., 1997 not be more appropriate references for 'the influence of seasonal recording'*

We have added Mix here as it is a good reference for the theory. Mulitza et al. (1997, Planktonic foraminifera as recorders of past surface-water stratification) deals more with the depth rather than seasonal effect so we will place this reference elsewhere.

*Pg. 3, Line 21: remove duplicate 'thus'*

Agreed.

*Pg. 5 line 3: change 'or' to 'including', as vital effects (the potential metabolic effects) are not exclusively inter-individual variation (given the individual life histories of foraminifera found within the sediment and or plankton tow samples.*

Agreed.

*Pg. 13 Line 9 'choose parameter values resembling this study'* but then state further *'these choices are partly arbitrary'* We will revise the parameter values for this example following the comments in SC1

*Pg. 15 line's 25-27 The scenario envisioned is performed by Lougheed et al. (2017)*

Reference added.

Once again, we thank you for your comments,

Andrew Dolman.

**References**

Anand, P., Elderfield, H. and Conte, M. H.: Calibration of Mg/Ca thermometry in planktonic foraminifera from a sediment trap time series, Paleoceanography, 18(2), 1050, doi:10.1029/2002PA000846, 2003.

Laepple, T. and Huybers, P.: Reconciling discrepancies between Uk37 and Mg/Ca reconstructions of Holocene marine temperature variability, Earth and Planetary Science Letters, 375, 418–429, doi:10.1016/j.epsl.2013.06.006, 2013.

Leduc, G., Sachs, J. P., Kawka, O. E. and Schneider, R. R.: Holocene changes in eastern equatorial Atlantic salinity as estimated by water isotopologues, Earth and Planetary Science Letters, 362, 151–162, doi:10.1016/j.epsl.2012.12.003, 2013.

Lohmann, G., Pfeiffer, M., Laepple, T., Leduc, G. and Kim, J.-H.: A model–data comparison of the Holocene global sea surface temperature evolution, Clim. Past, 9(4), 1807–1839, doi:10.5194/cp-9-1807-2013, 2013.

Lougheed, B. C., Metcalfe, B., Ninnemann, U. S. and Wacker, L.: Moving beyond the age-depth model paradigm in deep sea palaeoclimate archives: dual radiocarbon and stable isotope analysis on single foraminifera, Clim. Past Discuss., 2017, 1–16, doi:10.5194/cp-2017-119, 2017.

Marcott, S. A., Shakun, J. D., Clark, P. U. and Mix, A. C.: A Reconstruction of Regional and Global Temperature for the Past 11,300 Years, Science, 339(6124), 1198–1201, doi:10.1126/science.1228026, 2013.

Mix, A.: The oxygen-isotope record of glaciation, in North America and adjacent oceans during the last deglaciation., vol. K-3, pp. 111–135, Geological Society of America., 1987.

Mollenhauer, G., Eglinton, T. I., Ohkouchi, N., Schneider, R. R., Müller, P. J., Grootes, P. M. and Rullkötter, J.: Asynchronous alkenone and foraminifera records from the Benguela Upwelling System, Geochimica et Cosmochimica Acta, 67(12), 2157–2171, doi:10.1016/S0016-7037(03)00168-6, 2003.

Mulitza, S., Dürkoop, A., Hale, W., Wefer, G. and Niebler, H. S.: Planktonic foraminifera as recorders of past surface-water stratification, Geology, 25(4), 335–338, doi:10.1130/0091-7613(1997)025<0335:PFAROP>2.3.CO;2, 1997.

Müller, P. J., Kirst, G., Ruhland, G., von Storch, I. and Rosell-Melé, A.: Calibration of the alkenone paleotemperature index U37K' based on core-tops from the eastern South Atlantic and the global ocean (60 N-60 S), Geochim. Cosmochim. Acta, 62(10), 1757–1772, doi:10.1016/S0016-7037(98)00097-0, 1998.

Roche, D. M., Waelbroeck, C., Metcalfe, B. and Caley, T.: FAME (v1.0): a simple module to simulate the effect of planktonic foraminifer species-specific habitat on their oxygen isotopic content, Geosci. Model Dev. Discuss., 2017, 1–22, doi:10.5194/gmd-2017-251, 2017.

Scussolini, P., van Sebille, E. and Durgadoo, J. V.: Paleo Agulhas rings enter the subtropical gyre during the penultimate deglaciation, Climate of the Past, 9(6), 2631–2639, doi:10.5194/cp-9-2631-2013, 2013.

Shakun, J. D., Clark, P. U., He, F., Marcott, S. A., Mix, A. C., Liu, Z., Otto-Bliesner, B., Schmittner, A. and Bard, E.: Global warming preceded by increasing carbon dioxide concentrations during the last deglaciation, Nature, 484(7392), 49–54, doi:10.1038/nature10915, 2012.
* * *
[Figure]

[Figure]

**Fig. 1.** A comparison of dynamic and static habitat weights.

**Fig. 2.** Replacement for Fig. 4

---

## Author Comment (AC3) · 25 Jun 2018

Dear Paolo Scussolini,

Thank you for taking the time to read and comment on our discussion paper. We appreciate that in using your study as an example we should have made more effort to justify our choice of parameter values and to check their realism. Our main aim was to illustrate the capability of the sedproxy to simulate IFA type studies and to explore potential alternative explanations for patterns in paleo data. We should have stressed more clearly that we see bioturbation as an alternative explanation rather than presenting it as the most plausible explanation, which of course will depend heavily on the parametrisation.

Your suggestions will greatly improve the manuscript. Here we respond to your specific points in turn. We have included some relevant text from your comment in blue italicised text.

**1. Seasonality of d$^{18}$O at 400 m:**

*"First, Scussolini et al. (2013) analysed the planktic foraminifer Globorotalia truncatulinoides (sinistral coiling variety). This organism calcifies at depths beyond 400 or 600 m, according to the relevant literature and to Scussolini and Peeters (2013, Paleoceanography; doi: 10.1002/palo.20041; see also references therein), who compared values form core-top specimens to modern hydrography. At these depths, at the core site, there is hardly any seasonal variation in temperature and salinity. To assume 0.5 ‰ seasonal noise to mimic the del18O signal seems therefore inappropriate. I expect that this shouldn't change the position or magnitude of the peak in variability simulated by 'sedproxy', but it would be advisable to rectify the calculations to reflect this."*

We accept that there is very little seasonality in d$^{18}$O$_{sw}$ at these depths and we have removed seasonality from the calculation. However, we also see that we overlooked the much larger variation in d$^{18}$O over the depth habitat of Globorotalia truncatulinoides. Consequently, we have modified this example to demonstrate how the habitat weights can be used with a depth resolved rather than seasonally resolved input climate. We refer to Figure 2 of Scussolini Peeters (2013) to approximate the d$^{18}$O depth gradient and use a Gaussian distribution with mean of 520 m and standard deviation of 50 m for the habitat weights.

**2. Correction for instrumental variance measured on standards:**

*"Second, Scussolini et al. (2013) report that they 'corrected the variance of foraminiferal del18O by subtracting that of external calcite standards measured in the*

*same sequence', with the aspiration to clean their proxy from the spurious effect of measurement noise. It seems that Dolman and Laepple do not take this into account, as they 'assume a measurement noise of 0.1‰ del18O for the IFA and the bulk measurements'."*

Unfortunately, our description of the method was too brief as we did in-fact subtract this variance from the IFA variance estimated for the simulation output (line 484 in the supplementary .Rmd file). The effect on our Fig. 8 is however small, as the variance due to measurement error amounts to only 0.01 ‰$^2$.

**3. Speed of the climate transition:**

*"Third, Dolman and Laepple assumed 'a climate transition from 0.4 ‰ at 190 ka BP, to 2.6 ‰ at 90 ka BP'. The signal in core 64PE-174P13 goes from ca. 1.6 ‰ at 190 ka BP, to 1.3 ‰ at 90 ka BP (see fig. 2 in Scussolini et al. 2013). Where were the values of 0.4 and 2.6 ‰ taken from? In any case, this choice of such extended time frame is puzzling, as the sharp change in del18O occurs obviously across the glacial termination (ca. 140 to 125 ka BP)."*

Regarding the assumed climate transition between MIS 5 and 6, unfortunately the quoted 0.4‰ at 190 ka BP was a typographical error, the actual value used was 1.4‰ (it was correct in the code in Supplement 01). The upper value of 2.6‰ was taken from fig. 2 in Scussolini et al. 2013 as the approximate mean value prior to the transition at around 140 ka BP.

Additionally, we could have described the logistic function more precisely. The end points of the function were set at 190 ka BP (1.4‰ and 90 ka BP (2.6‰ but most of the transition occurs during a much shorter window between about 130 and 135 ka BP. We will improve the description in the revised version.

**4. Bioturbation depth:**

*"Further, assuming bioturbation reaching 10 cm from the top of the sediment will obviously produce a peak in variability in any record across a signal transition such as a glacial termination. While it is unrealistic to think that bioturbation is absent from core 64PE-174P13, Scussolini et al. (2013) advanced multiple lines of reasoning to exclude strong bioturbation in core 64PE-174P13, not least visible laminations in parts of the record (see also the author's response to referee 1, who raised specifically the point of bioturbation: https://www.clim-past-discuss.net/9/C511/2013/cpd-9-C511-2013.pdf). An additional argument against the role of bioturbation and in favor of an interpretation of the variability signal as proxy for Agulhas rings comes from Scussolini et al. (2015, Geology, doi: 10.1130/G36238.1). There, a tight coupling is shown between the Agulhas rings proxy with the ice-volume-corrected seawater del18O of G. truncatulinoides, a proxy for the high salinity anomalies that Agulhas rings seem to have introduced at the core location (see below a snapshot of the relevant figure in Scussolini et al. 2015, showing the two proxies). It is important to note that the two proxies are analytically independent of one another. It is not clear from the manuscript whether the authors have reasons to prefer the interpretation of the signal in terms of bioturbation."*

We accept this point. The plausibility of bioturbation as an explanation for the variance peak will depend strongly on the bioturbation depth, which is poorly constrained.

We have re-run these simulations using a range of bioturbation depths and using the depth-resolved input climate and habitat weights mention above in place of seasonality. While the peak in variance remains clear down to bioturbation depths as low as 3 cm, the absolute value and width of the variance peak are a little lower than that seen in Fig. 2 of Scussolini et al. 2013 (see Fig.1). At the same time, for bioturbation depths of 3 and 5 cm, the apparent speed of the climate transition is consistent with the sharpness of transition (approximately 8 ka) seen in the bulk record for G. truncatulinoides in Fig. 2. of Scussolini et al. 2013 (see Fig.2). However, for 10 cm of bioturbation the transition is too spread out.

We cannot of course exclude enhanced Agulhas leakage as the source of increased

IFA variance across the MIS 5-6 transition, and as noted there is other evidence for increased leakage such as the tight coupling between the Agulhas rings proxy and the $d^{18}O$ of G. truncatulinoides. However, given that bioturbation depths as low as 3 cm still produce a quite visible variance peak we think that bioturbation is at least a plausible mechanism behind some of the change in variance over the MIS 5-6 transition. We will modify the manuscript to improve the description of the simulation, to describe the use of depth rather than seasonal weighting, and to make clear that we see bioturbation as a possible alternative mechanism but that this depends heavily on the parametrisation.

Once again, we thank you for your comments,

Regards,

Andrew Dolman.

**Fig. 1.** IFA variance for different bioturbation depths.

[Figure]

**Fig. 2.** Simulated bullk and IFA proxies for different bioturbation depths.